# Learning Context-Adapted Video-Text Retrieval by Attending to User Comments

## Abstract

Learning strong representations for multi-modal retrieval is an important problem for many applications, such as recommendation and search. Current benchmarks and even datasets are often manually constructed and consist of mostly clean samples where all modalities are well-correlated with the content. Thus, current video-text retrieval literature largely focuses on video titles or audio transcripts, while ignoring user comments, since users often tend to discuss topics only vaguely related to the video. In this paper we present a novel method that learns meaningful representations from videos, titles and comments, which are abundant on the internet. Due to the nature of user comments, we introduce an attention-based mechanism that allows the model to disregard text with irrelevant content. In our experiments, we demonstrate that, by using comments, our method is able to learn better, more contextualised, representations, while also achieving competitive results on standard video-text retrieval benchmarks.

## 1 Introduction

Training large scale multi-modal models from paired visual/text data from the web has seen great success in video understanding and retrieval. However, typically only the caption or "alt text" is used during training, ignoring potentially relevant text present on the web page, such as user comments, tags, descriptions and other metadata.

The challenge in using this extra context is that it may often be not directly relevant to the video (a comment "cool video!"), or it may be relevant but non-distinctive (a tag "cat" could apply to many videos). Similarly, other modalities can also exhibit this behavior. For example, current work that learns from audio-visual correspondence (Asano et al., 2020; Alwassel et al., 2020; Morgado et al., 2020) need clean datasets such as VGGSound (Chen et al., 2020a) to learn meaningful correspondence between videos and sound, whereas online videos tend to have for example background music that replaces the actual sounds happening in the video.

In this paper, we propose a method that can take advantage of this auxiliary context while simultaneously filtering it for meaningful information. Most current models enforce a strict correlation between the different input modalities under the assumption that all are informative of the content. The main intuition of our work is that when training a model on partially unrelated data, we need to introduce a mechanism that allows the model to discount auxiliary data when it is not helpful for the task.

To this end, we build a model with a hierarchical attention structure. Current representation learning models that are based on transformer architectures already exploit the idea of an attention mechanism to model the correlation between different *parts* of an input signal. For example in text understanding, the attention mechanism is applied per word, allowing the model to understand the structure of natural language. Even though in principle one could use the same scheme to model the importance of different text inputs on a per-word basis, we find that this makes it difficult to learn the individual importance of inputs. Moreover, due to the computational complexity of current transformers (squared with sequence length) this approach would only work for a small number of short comments. We thus add a second layer of attention per processed input that allows the model to assess the amount of information at a higher level of features. We find that this mechanism aligns well with the intuition that some inputs are very relevant to the problem and others can be disregarded.

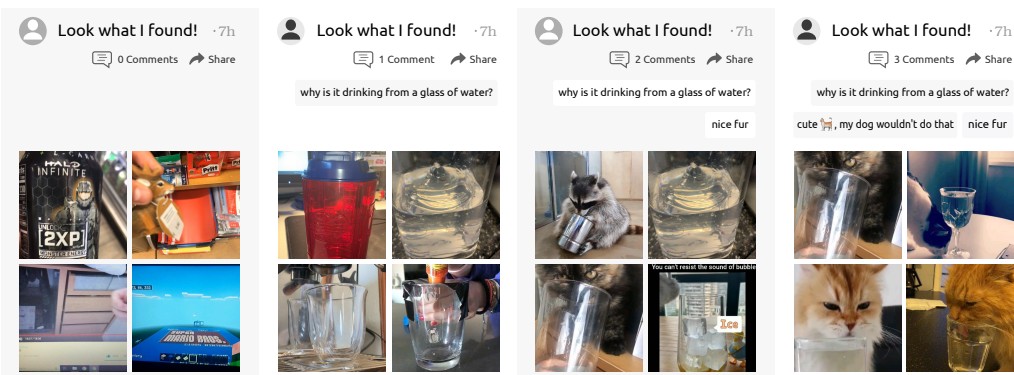

Figure 1: **Video Retrieval from Title and comments.** We show the top 4 videos retrieved for the ambiguous title "Look what I found!", and from left to right we progressively add more comments which our model uses to refine the results.

In our experiments we show that we can indeed learn meaningful information from user comments that translates to good zero-shot retrieval performance on common benchmarks. Additionally, we can show that the model can correctly identify whether auxiliary information is informative of the content of a video or not.

The ability to incorporate auxiliary contextual information also opens up possibilities for useful applications. In the video retrieval setting our method can be used to iteratively refine a text descriptor with new inputs as shown in Fig. 1, allowing incremental searching. In the zero-shot video classification setting (*i.e.*, "retrieving" the correct class description prompt) the prediction for an ambiguous video can be steered towards the correct class using surrounding text from a webpage or user hints (Fig. 4).

## 2  RELATED WORK

In this work, we focus on multi-modal learning with a particular focus on learning video-text encoders for retrieval by proposing a novel, multi-modal adaptation module.

**Video-text pretraining.** Originating from the NLP domain, where the transformer architectures has been a key ingredient and subject to optimization in a multitude of ways (Vaswani et al., 2017; Devlin et al., 2019; Radford et al., 2019; Raffel et al., 2019; Lewis et al., 2020b; Clark et al., 2020; Lewis et al., 2020a), it has recently found applications in the vision-language domain. For example, recent works have leveraged transformers to learn generalizeable image (Desai & Johnson, 2020; Sariyildiz et al., 2020; Chen et al., 2020b) or multi-modal image-text (Li et al., 2019; Lu et al., 2019; Tan & Bansal, 2019; Su et al., 2019; Li et al., 2020; Chen et al., 2019) and video-multilingual text (Huang et al., 2021) representations. A few works (Sun et al., 2019b;a; Zhu & Yang, 2020; Luo et al., 2020) combine visual and text modalities as inputs to a BERT model to simultaneously learn semantic video and text representations. For representation learning, the availability of large-scale datasets such as HowTo100M (Miech et al., 2019) has enabled more effective pretraining of video-text representations for multiple downstream tasks. More recently, Patrick et al. (2020b) show that adding a generative objective to contrastive pretraining can yield gains in video-text downstream tasks. Based on the CLIP model (Radford et al., 2021), which works well even without finetuning for some retrieval tasks (Luo et al., 2021), Bain et al. (2021) train video-text CLIP-initialized models by gradually scaling up video training from image training and a custom dataset. While we also start with a CLIP initialization as in Luo et al. (2021), the focus of our paper lies in developing a novel method for leveraging user comments, a modality that has previously been overlooked in the text-video retrieval literature, as a valuable source of information. As is standard, the pretrained representations are subsequently evaluated on smaller datasets such as MSVD (Venugopalan et al., 2015) and MSR-VTT (Xu et al., 2016a).

**Multi-modal domain adaptation.** While residual adapters for domain adaptation have been explored for uni-modal models such as CNNs, e.g. in Rebuffi et al. (2017), there are no works that translate

this concept to the multi-modal domain, where cross-modal learning dominates (Alwassel et al., 2020; Asano et al., 2020; Morgado et al., 2020).

**Multi-modal pretraining.** There is little prior work using user comments as additional context in multimodal settings. The use of user comments and reactions to refine predictions has been discussed by Halevy et al. (2020) for minimising harms on facebook.com. Overall, we are the first to demonstrate that user comments can be used as a complementary modality when learning video-text representations.

## 3 METHODS

In this section we will first recap the mechanism behind current contrastive, multi-modal representation learning methods that rely on clean data. We will then introduce our Context Adapter Module that allows learning from the auxiliary modality through an attention mechanism. Finally, we will describe how we can extend an existing backbone for images to videos, to be able to leverage large, pretrained models.

### 3.1 BACKGROUND

In multi-modal representation learning we are given a dataset $\mathcal{X}$ of $N$ samples $x_i \in \mathcal{X}, i \in \{1, \ldots N\}$ that individually consists of different signals. Most previous work focuses on two modalities and we will—for now—also adhere to this standard to simplify the notation. This means that each input sample $x_i = (v_i, t_i)$ is a pair of—in our case—a visual input $v_i \in \mathcal{V}$ and its associated text, often the title, $t_i \in \mathcal{T}, 1 \le i \le N$.

The goal is now to learn mappings $f_v : \mathcal{V} \mapsto \mathcal{Y}, f_v(v_i) = \phi_{v,i}$ and $f_t : \mathcal{T} \mapsto \mathcal{Y}, f_t(t_i) = \phi_{t,i}$ from each of the modalities to a $d$-dimensional, joint embedding space $\mathcal{Y} = \mathbb{R}^d$. Recent methods, such as Radford et al. (2021), learn the mapping (in their case from images and their captions) to the embedding space with a double contrastive loss over a mini-batch $\mathcal{B} \subset \mathcal{X}$ using an affinity matrix $A$ computed between all pairs of samples in the batch:

$$A_{ij} = \left\langle \frac{\phi_{v,i}}{\sqrt{\tau}\|\phi_{v,i}\|}, \frac{\phi_{t,j}}{\sqrt{\tau}\|\phi_{t,j}\|} \right\rangle \tag{1}$$

An entry $A_{ij}$ measures the similarity between the embeddings $\phi_v(v_i)$ and $\phi_t(t_i)$ via cosine similarity that is scaled by a temperature parameter $\tau$. The idea is now to maximize the similarity between the embeddings from the *same* sample, *i.e.* the diagonal of $A$ and minimize all non-diagonal entries. This can be achieved efficiently using a double-contrastive formulation that operates across columns and rows of $A$,

$$\mathcal{L}(A) = \frac{1}{2} \sum_{i=1}^{|\mathcal{B}|} \frac{A_{ii}}{\log \sum_{j=1}^{|\mathcal{B}|} \exp A_{ij}} + \frac{A_{ii}}{\log \sum_{j=1}^{|\mathcal{B}|} \exp A_{ji}}. \tag{2}$$

This formulation has the neat effect that it accomplishes maximizing the diagonal entries and minimizing all other entries of $A$ in one self-balancing formulation. However, it makes the critical assumption that both modalities are equally informative of each other. In the case of sometimes irrelevant data, or when one modality has much less information content than the other (*e.g.* "*nice video!*"), this assumption does not hold and training with this objective will result in a very volatile learning objective and thus a sub-optimal joint embedding.

In the next section we will introduce our Context Adapter Module that is able to deal with this type of inputs by allowing it to discount information when it is not relevant for the context.

### 3.2 CONTEXT ADAPTER MODULE

In order to capture and filter the relevant information from the comments, we propose a transformer-based Context Adapter Module (CAM) which operates in a residual fashion, additively adapting either the visual or text branch of CLIP with contextual information obtained from the comments (see Fig. 2). Formally, we are now adding another modality—the comments—to the input which

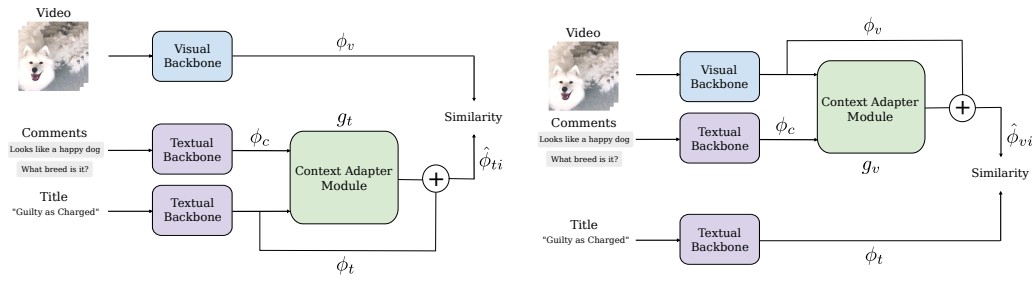

(a) Adapting the text branch.      (b) Adapting the visual branch.

Figure 2: **Method Overview.** We introduce a context adapter module that uses inputs of the auxiliary modality to adapt the embedding of another branch. With this module the model is able to accept or discount information.

extends it to $x_i = (v_i, t_i, c_{i,1}, \ldots, c_{i,M})$ with $c_{i,k} \in \mathcal{T}$. To reduce clutter in the notation, we have defined a fixed number of comments $M$ for each sample. Since both, title and comments, share the same underlying modality, namely text, we can leverage the same encoder to transform comments to embeddings $f_t(c_{ik}) = \phi_{c,ik}$.

As we expect the comments to be sometimes unrelated, our Context Adapter Module needs a mechanism to discount off-topic comments and update the primary modality $\phi_v(v_i)$ or $\phi_t(t_i)$, steering it in the most informative direction.

We introduce this mechanism as a function of both the primary modality and the comment embeddings $\phi_{c,ik}$, as we want to compare the informativeness of all these inputs at a high level. To this end, we design adapter modules $g_v$ and $g_t$ that extract information from the comments in the form of a residual:

$$\hat{\phi}_{oi} = \phi_{oi} + g_o(\phi_{oi}, \phi_{c,i,1}, \ldots, \phi_{c,i,M}), \, o \in \{v, t\} \tag{3}$$

With the adapted embeddings $\hat{\phi}_{vi}$ and $\hat{\phi}_{ti}$ we recompute the affinity matrix (now $\hat{A}$) (Eq. 1) and use it for the loss $\mathcal{L}(\hat{A})$. This design has several advantages. On one hand, extracting "only" a residual from the auxiliary inputs $c_{ik}$ means that the model is easily able to ignore them by predicting $g(\cdot) = 0$. On the other hand, this effectively allows us to skip the adapter module when we benchmark the model on a dataset that does not have comments, while still learning the joint embedding from richer data.

In practice, we implement $g$ as a small transformer architecture. Rather than operating on tokenised words, this transformer operates on embeddings ($\phi_{vi}$ and $\phi_{ti}$) themselves, taking as input the encoded feature from the branch to be adapted, along with comment features $\phi_{c,ik}$. By treating embeddings as tokens in their own right, we allow the embeddings to attend to each other and learn what combinations of the inputs should be used to update the original feature through the residual connection.

Additionally, to avoid bleeding information between the two modalities through the Context Adapter Module, during training, we only adapt either the video embedding with $g_v$ or the text embedding with $g_t$. If we would use both adapters simultaneously, there is a trivial solution that minimizes the loss $\mathcal{L}$: when the adapters learn to remove the original embedding through the residual, *e.g.* $g_o(\phi_{oi}, \{\phi_{c,i,k}, \}) = -\phi_{oi} + \phi_{c,i,1}$ both adapted embeddings become the same $\hat{\phi}_{vi} = \hat{\phi}_{ti}$ which trivially maximizes their similarity, thus preventing the model to learn a meaningful modality alignment. To prevent the model from learning a transformation of the embedding space through the residual, we train only one adapter at a time. We also consider the case where $g_v = g_t$ and choose a random modality to adapt for each minibatch, leading to an adapter that is agnostic to the source of the primary embedding.

## 3.3 FROM IMAGES TO VIDEOS

To leverage the capacity of large pre-trained computer vision models, we adopt the architecture by Radford et al. (2021) as our backbone models $f_v$ and $f_t$. While this transformer was trained on a huge volume of image-text data, it cannot be applied directly to videos since it is built for images and has no temporal extent. To take advantage of the temporal information present in video

Table 1: **Comments as a modality.** Treating comments as an auxiliary modality to a video improves Text-to-Video retrieval for both RedditVC and KineticsComments test sets. As a baseline, we use the CLIP model on just the first frame. We then train just our CAM on top of CLIP, adapting the visual embedding with information from the comments, which improves retrieval metrics. Training our full model, which includes temporal attention, on RedditVC again validates that the inclusion of comments helps, with a further jump in Kinetics performance when using the Kinetics training set.

| Method | #Frames | Train w/ Comm. | Test w/ Comm. | Reddit Eval. | | Kinetics Eval. | |
|---|---|---|---|---|---|---|---|
| | | | | R@5 | R@10 | R@5 | R@10 |
| CLIP | 1 | no | no | 7.29 | 9.47 | 36.6 | 47.0 |
| Ours + Our CAM (RedditVC) | 1 | yes | no | 7.41 | 9.72 | - | - |
| Ours + Our CAM (RedditVC) | 1 | yes | yes | 13.95 | 18.24 | 42.4 | 53.3 |
| Ours (RedditVC) | 8 | yes | no | 7.73 | 10.28 | 43.4 | 55.6 |
| Ours (RedditVC) | 8 | yes | yes | 11.15 | 14.89 | 46.3 | 58.6 |
| Ours (RedditVC+KineticsCmts) | 8 | yes | no | 7.11 | 9.38 | 50.3 | 63.7 |
| Ours (RedditVC+KineticsCmts) | 8 | yes | yes | 10.93 | 14.58 | 52.9 | 67.4 |

data, we use the Divided Space-Time attention mechanism recently introduced in the TimeSformer architecture (Bertasius et al., 2021). We modify the image transformer architecture by adding patchwise self-attention across 8 frames in time to each of the 12 residual attention blocks, followed in each case by a zero-initialised linear layer. We also add a learned temporal position embedding which is summed to the input and again zero-initialised. The initialisation is transparent, such that when loading pretrained weights trained from images, at initialization time, the modifications do not affect the inference of the model. During training, the model can then gradually activate the additional temporal components to learn from the temporal information of a video. Full details on the architecture are provided in the appendix.

## 4 EXPERIMENTS

### 4.1 IMPLEMENTATION DETAILS

We use CLIP (Radford et al., 2021) as the initialisation for the backbone. Our concrete implementation of the CAM $g$ is a 2-layer transformer, consisting of two residual multihead self-attention blocks. The input consists of $M + 1$ input embeddings (for the $M$ comments and title/video embedding $\phi_{oi}$) having 512 dimensions each. Each block performs 8-head self-attention on the inputs, followed by two linear layers with output size 2048 and 512 respectively. LayerNorm normalisation is used, along with GELU activation following the first linear layer. From the $M + 1$ outputs of the transformer, we take only the output at index 1 (the index of the text/video embedding), and pass it through a final $512 \times 512$ linear layer. We use learning rate 0.001 for training the CAM, and $1 \times 10^{-6}$ when training the entire model, using the Adam Kingma & Ba (2015) optimizer.

All implementation and architecture details can be found in the appendix.

### 4.2 PRE-TRAINING DATASETS

**RedditVC.** We collect a dataset "RedditVC" of videos along with their titles and comment threads from social news site reddit.com, using their provided API. The videos are collected and used in a manner compatible with national regulations on usage of data for research. Unlike most curated video datasets, this data is more representative of the types of videos shared "in the wild", containing a large proportion of videogames, screenshots and memes.

Using a classifier trained on a small amount of labelled data, we estimate that videogame footage makes up 25% of examples, other screenshots, memes and comics make up 24%, live action footage is 49% and artistic styled content (such as drawn animation) is 2%. The average video length is 33s.

From 1 million raw videos collected, we perform deduplication and filtering, ending up with a training set of 461k videos, and validation and test sets of 66k videos each. For the video evaluation results in Table 1, we use a subset of the test set, consisting of 5000 videos with at least three comments each.

Table 2: **Ablation Study.** Comparing Text-to-Video and Video-to-Text retrieval results between different baselines and our method.

| | Finetune Backbone | Train. CAM | Eval. CAM | Text $\rightarrow$ Video R@1 | R@10 | Video $\rightarrow$ Text R@1 | R@10 |
|---|---|---|---|---|---|---|---|
| (a) | ✗ | - | - | 1.70 | 7.30 | 1.57 | 6.86 |
| (b) | ✓ | - | - | 2.25 | 9.85 | 2.22 | 9.28 |
| (c) | ✓ | $\text{rand}(\phi_t, \phi_{ck})$ | - | 2.06 | 9.40 | 2.04 | 8.83 |
| (d) | ✓ | $\text{rand}(\phi_t, \phi_{ck})$ | $\text{avg}(\phi_t, \phi_{ck})$ | 1.64 | 7.99 | 1.96 | 9.49 |
| (e) | ✓ | $\text{avg}(\phi_t, \phi_{ck})$ | - | 1.67 | 7.79 | 1.97 | 9.01 |
| (f) | ✓ | $\text{avg}(\phi_t, \phi_{ck})$ | $\text{avg}(\phi_t, \phi_{ck})$ | 2.49 | 11.75 | 2.63 | 12.06 |
| (g) | ✓ | $g_t$ | - | 2.28 | 9.70 | 2.14 | 9.22 |
| (h) | ✗ | $g_t$ | $g_t$ | 1.98 | 9.93 | 1.70 | 8.87 |
| (i) | ✓ | $g_t$ | $g_t$ | 2.06 | 10.49 | 1.96 | 9.98 |
| (j) | ✓ | $g_v$ | - | 2.22 | 9.67 | 2.11 | 9.11 |
| (k) | ✗ | $g_v$ | $g_v$ | 2.71 | 12.22 | 2.86 | 12.44 |
| (l) | ✓ | $g_v$ | $g_v$ | 2.86 | 13.07 | 2.93 | 13.07 |

In addition to comments, for one experiment in Table 4 we also obtain textual labels of objects in the video thumbnails, using the Google Vision API. We obtain labels for 10,212 different classes, from "Abacus" to "Zwiebelkuchen". This serves as a useful comparison, giving textual metadata that can be expected to be more directly related to the visual video contents than user comments, and can be seen as a proxy for user-generated "hashtags". It also illustrates how, by using text as an intermediary, we can integrate predictions from any black-box classifier in the same framework.

**KineticsComments.** As an additional video dataset with comments, we construct a dataset based on Kinetics-700 (Kay et al., 2017; Carreira et al., 2019), for which we download the videos along with associated YouTube metadata including title, description and comments. We translate non-English titles and descriptions into English using a commercial translation API. We use the title as the primary text modality, and for auxiliary context we use comments and (when training) sentences from the description. This leaves us with 484,914 videos for training, each being around 10s. We also construct a test set, consisting of videos from the Kinetics test set for which we have at least 3 comments, giving a set of 2700 videos which we use to evaluate our method in Table 1.

### 4.3 EVALUATING THE CONTEXT ADAPTER MODULE

We evaluate our Context Adapter Module on the above described datasets with comments in Table 1. We find that on both datasets adding comments boosts the retrieval performance significantly, confirming the value of the modality. Further, moving from images to videos also improves the performance. Since KineticsComments (human actions) is quite different to RedditVC (broad range of videos, games, etc.) combining both datasets bridges the domain gap and improves the evaluation performance on KineticsComments.

**Baseline Comparisons.** Since the main purpose of the Context Adapter Module is to extract meaningful information from sometimes unrelated auxiliary data, we perform most of the experiments in this section with an image/text backbone using a single video frame instead of a video/text backbone to reduce the computational burden of the evaluation.

In Table 2 we evaluate the efficacy of the proposed Context Adapter Module against various baselines in various settings on our RedditVC dataset. The most basic baseline is achieved by not using any form of context adaption, which degenerates the model to the one proposed by Radford et al. (2021).

Across all experimental settings, we find that finetuning the backbone architecture helps to improve the performance, while using the proposed CAM with frozen backbones (rows h,k) even outperforms the naive baseline of row (a). We also compare to two baselines. The first baseline is randomly replacing the title features with comments during training (Table 2(c-d)), however, this does not yield any improvement over no adaptation. In the second baseline we average the title features $\phi_t$ and the

Table 3: **Adapting different heads.** Text-to-video R@10 results on RedditVC-5. We either adapt the text or image features.

| train CAM | test $\phi_v$ | test $\phi_t$ |
|---|---|---|
| $g_v$ | 11.48 | 2.98 |
| $g_t$ | 0.50 | 9.48 |
| $g_v$ & $g_t$ | 10.08 | 9.16 |

Table 4: **Influence of Auxiliary Data.** Text-to-video retrieval results on RedditVC-5. We vary the auxiliary text source during training and testing. Using images labels from a classifier during training works well but does not generalize.

| | R@10 | | |
| Aux. Train Data | Im. Labels | Comments | Rand. Words |
|---|---|---|---|
| Image labels | 22.20 | 6.91 | 3.06 |
| Comments | 10.45 | 9.48 | 6.67 |

comment features $\phi_c$ which can be seen as a very simple adapter module (Table 2(e-f)). This yields a small benefit when both training and evaluation use averaging, but falls behind utilizing proposed CAM on the visual modality (Table 2(l)). Surprisingly, we find that training with comments performs best when they are added to the image branch (rows j-l), as opposed to the text-branch (rows g-i). We hypothesize that this is because the comments can better adapt the semantically richer feature of the visual modality as opposed to the text, which can often be as short as a single word.

**Adapting Different Modalities.** Since both the visual embedding $\phi_v$ and the text embedding $\phi_t$, come from the same embedding space and should be similar for the same sample, we perform an experiment where we swap the embeddings during evaluation time. In Table 3 we find that there are still differences between $\phi_v$ and $\phi_t$ and swapping results in decreased performance. When we train both adapters we can achieve good performance in both cases.

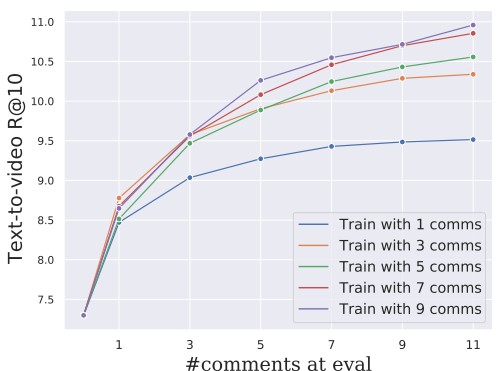

Figure 3: **Varying Number of Comments.** The model can effectively benefit from additional auxiliary information and models trained with more comments become better at extracting information even when fewer are available during testing.

Table 5: **MSVD** (Chen & Dolan, 2011) Zero-shot text-to-video retrieval. CLIP is an important baseline since we use it as our backbone initialisation. Even though comments are not available for this benchmark the model trained with comments has learned better representations.

| Method | R@1 | @5 | @10 | Rk |
|---|---|---|---|---|
| CLIP-Straight (Portillo-Quintero et al., 2021) | 37.0 | 64.1 | 73.8 | 3 |
| CLIP4Clip (Luo et al., 2021) | 38.5 | 66.9 | 76.8 | **2** |
| Ours (no comments) | 39.5 | 68.4 | 78.3 | 2 |
| Ours (no TimeSformer) | 37.7 | 66.0 | 76.5 | 3 |
| Ours | 39.5 | 68.6 | 78.4 | 2 |
| Ours (with Kinetics) | **40.9** | **69.5** | **78.7** | 2 |

**Auxiliary Information.** Additionally, Table 4 shows how varying the source of the text at train and test time impacts the retrieval results. Whilst the performance of the model trained with image labels degrades when given comments at evaluation time, a model trained with comments still benefits from image labels at test time. As expected, the performance of both models is negatively impacted when tested with uncorrelated random words as additional text. However, the model trained with image labels does noticeably worse in this setting. This suggests that it has become overly reliant on image labels and easily fooled by irrelevant information, which overrides the title feature.

**Varying the number of comments.** In Fig. 3 we vary the number of comments during training and evaluation time. Training on more comments results in better retrieval results when evaluated on the same number of comments. However, while there is little difference for models trained with only a few comments, we found that increasing the number of comments at train time leads to models that are more robust to different number of comments at test time, indicating that training with more

comments learns an overall better context adapter module that is then able to work efficiently in various settings.

## 4.4 COMPARISON TO THE STATE OF THE ART

In order to compare to other methods, we use the text-to-video retrieval benchmarks MSR-VTT (Xu et al., 2016a) and MSVD (Venugopalan et al., 2015). While these benchmarks do not directly take into account the novel ability of our method to consider contextual information, they give a measure of the transferability of the learned text and video features. For evaluation, we skip the CAM and use $\phi_{vi}$ and $\phi_{ti}$ directly since these benchmarks do not involve comments. Fine-tuning a learned representation only yields an indirect measure of the original model's quality. Since we are interested in the immediate quality of the learned representations, we focus this evaluation on zero-shot performance on the datasets, without fine-tuning. Nonetheless, fine-tuning results can be found in the appendix.

**Datasets.** For MSVD (Table 5), we use the test split consisting of 670 videos and approximately 40 captions each.

For MSR-VTT (Table 6), we use the 1k-A or "jsfusion" evaluation split of the data, consisting of 1000 (video,text) pairs and 20 captions per video.

Moreover, we show the zero-shot performance of our method on 2 additional datasets. In Table 7 we report the Text-to-Video retrieval results on the ActivityNet (Fabian Caba Heilbron & Niebles, 2015) dataset. We evaluate on the "val 1" split, as in SSB (Patrick et al., 2020b) who we compare to. In Table 8, we show our Text-to-Video retrieval performance on the LSMDC Rohrbach et al. (2017) dataset, where we use the test set comprising of 1000 videos.

Finally, we report the Video-to-Text retrieval results for MSVD and MSR-VTT, which follow similar patterns, in the Appendix.

**Results.** Across all benchmarks we improve over all other methods in zero-shot retrieval performance. On MSR-VTT our performance is even slightly better than CLIP4Clip (Luo et al., 2021), that uses the same backbone initialization as us but additionally trains on HowTo100M which is more than 100 times larger than our dataset.

It is interesting to note, that we are the only method that evaluates zero-shot performance on such a wide spectrum of datasets which shows good generalization performance of the learned representations.

**Using comments, videos and more data.** In Tables 6, 9 and 10 we also show the performance of our model in different variants.

Across all three benchmarks, our model trained with comments is consistently better than the variant trained without comments. This means that even though the evaluation is performed without comments (none of these datasets provides user comments), the additional context during training has improved the underlying video-text representations.

Similarly, using video instead of a single image improves the results as well. The main difference here, however, is that the video is also available during evaluation. Thus, the observation is that video provides more or a better context for retrieval than single images.

Finally, we find that adding Kinetics to the training set has only a small effect on the quality of the learned representation and can—depending on the domain gap between Kinetics and the benchmark dataset—even affect the results slightly negatively.

Thus, while achieving state-of-the-art performance, we can also show that comments provide a strong signal to improve multi-modal representation learning, even when comments are not available during test time.

Table 6: Text-to-video retrieval on MSR-VTT. **Vis. Enc. Init.:** Datasets used for pretraining visual encoders for tasks *other than visual-text retrieval*, eg object classification. **pre-training:** Visual-text pretraining data. † Object, Motion, Face, Scene, Speech, OCR and Sound classification features. **#samp. PT:** size of the pre-training dataset in millions of samples.

| Method | Vis Enc. Init. | pre-training | #samp. PT | R@1 | R@5 | R@10 | Rk |
|---|---|---|---|---|---|---|---|
| HT MIL-NCE (Miech et al., 2019) | - | HowTo100M | 100M | 7.5 | 21.2 | 29.6 | 38 |
| SupportSet (Patrick et al., 2020b) | IG65M, ImageNet | HowTo100M | 100M | 8.7 | 23.0 | 31.1 | 31 |
| FiT (Bain et al., 2021) | ImageNet | CC3M, WebVid-2M | 5.5M | 18.7 | 39.5 | 51.6 | 10 |
| CLIP4Clip | - | CLIP, HowTo100M | 500M | 32.0 | **57.0** | 66.9 | 4 |
| VideoCLIP (Xu et al., 2021) | - | HowTo100M | 100M | 10.4 | 22.2 | 30.0 | - |
| Ours - (no comments) | - | CLIP, RedditVC | 401M | 31.5 | 54.3 | 64.5 | 4 |
| Ours - (no TimeSformer) | - | CLIP, RedditVC | 401M | 31.9 | 53.5 | 65.3 | 5 |
| Ours | - | CLIP, RedditVC | 401M | **32.2** | 54.9 | 65.4 | 4 |
| Ours - (with K700) | - | CLIP, RedditVC, K700 | 402M | 30.8 | 56.1 | **67.1** | 4 |

Table 7: **ActivityNet** (Fabian Caba Heilbron & Niebles, 2015) Text-to-Video retrieval.

| Method | R@1 | R@5 | R@10 | MedR |
|---|---|---|---|---|
| SSB (Patrick et al., 2020b) | 0.0 | 0.2 | 0.3 | 2238 |
| Ours | 8.4 | 22.7 | 33.0 | 26 |

Table 8: **LSMDC** (Rohrbach et al., 2017) Text-to-Video retrieval.

| Method | R@1 | R@5 | R@10 | MedR |
|---|---|---|---|---|
| JSFusion (Yu et al., 2018) | 9.1 | 21.2 | 34.1 | 36.0 |
| CE (Liu et al., 2019) | 11.2 | 26.9 | 34.8 | 25.3 |
| MMT (Patrick et al., 2020a) | 12.9 | 29.2 | 38.8 | 19.3 |
| FiT (Bain et al., 2021) | 15.0 | 30.8 | 39.8 | 20.0 |
| Ours | 24.0 | 48.0 | 60.1 | 6 |

## 4.5 LIMITATIONS

We find that the context adapter can be led to override the information in a title if we adversarially craft comments that all point to different content. Qualitative examples of this can be seen in Fig. 4. The model without comments, correctly associates the image with a cookie (jar), however when adding a comment about a "dog" the model prefers the dog label over the cookie label. More examples can be found in the appendix (Appendix A.5 and the results browser in the supplementary material).

## 5 CONCLUSION

We have presented the Context Adapter Module, which is able to extract information from auxiliary input sources for learning a joint, multimodal embedding. In our experiments, we are able to show that the model improves the representation quality of video-text embedding learning when adapting the representation with user

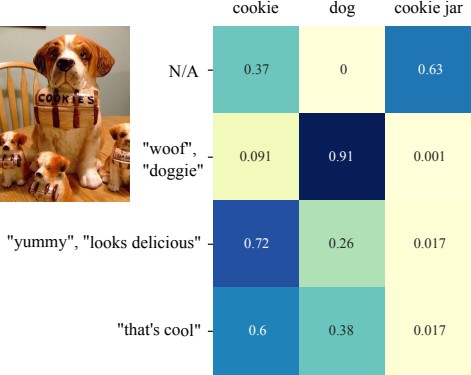

Figure 4: **Failure Case.** A heatmap showing the similarities between the image adapted with different comments (rows), and captions (columns). The adapter can steer away the embedding from the right association "cookie jar" depending on the comment input.

comments. Moreover, it is able to identify whether an auxiliary input is relevant to the content in the other modalities or not. This mechanism could, for example, be used to filter datasets for meaningful auxiliary content. The model can be initialized from existing, large-scale image-text backbones and the context adapter learns from a comparatively small number of examples (<500k). The module itself is modality-agnostic and can potentially be used in a large variety of multi-modality representation learning tasks such as audio-visual learning, or to combine the outputs of unreliable expert models.

REPRODUCIBILITY AND BROADER IMPACT

Online data collected from user-generated content are subject to all sorts of biases and potentially problematic content. We have carefully selected the discussion topics from which we source the data, to include topics with a lower chance of problematic topics such as pets, art, jokes and games. On top of that we filter text content with Detoxify (Hanu & Unitary team, 2020) for harmful content. The use of a content filtering text model such as Detoxify can introduce its own biases. Although we have used the 'unbiased' version of Detoxify to minimise these, there might still be particular words that are associated with offensive or hateful language that will likely be strongly correlated with high toxicity scores, regardless of the intent of the author. This could lead to overfiltering non-toxic text that contains particular identity terms, which might result in discrimination against already vulnerable groups. Moreover, the user-base of Reddit is predominantly young, male and western (Barthel et al., 2016) and thus inherently there will be certain biases present regardless of how thoroughly the dataset is filtered. The use of a primarily English language data source and pretrained model means learned concepts will be far from globally representative. The social biases discussed in Section 7.1 of Radford et al. (2021) all still apply. Additionally, as the data is downloaded from a public website, content creators and users could not be asked for consent to be included in the dataset.

The presented context adapter mechanism has a wide applicability across different multi-modal tasks where one input source is expected to sometimes be uninformative. However, as detailed in the previous section, when trained on online data, the model has no mechanism to distinguish between true and false statements and will simply learn from correlations. It has been repeatedly shown that algorithms trained on online user content are often unfit for any kind of production system and pose risks (Bender et al., 2021). Our work and analysis is for research purposes only and we strongly advise against using models trained on the RedditVC dataset in non-research applications.

Given the above discussion on impact and biases, we refrain for now from publicly releasing the RedditVC dataset or models trained on it, as their intent is for research proposes only and not fit for real-world applications. However, we are planning to release the comments collected for the KineticsComments dataset as the content of the videos is much more constrained, which results in easier filtering of the data.

For further reproducibility of the general approach we will release the code. Additionally, models trained on datasets other than RedditVC will be released as well.

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

# A APPENDIX

## A.1 QUALITATIVE EXAMPLES

In Fig. 5 we adapt the text branch, similar to Fig. 1 of the main paper. The example in the second row of Fig. 5 shows how our Context Adapter Module can leverage the comments to learn that the content is indeed about parrots, as opposed to dogs. The fourth row shows that without comments, the title alone can be extremely ambiguous while comments can again guide the model to retrieve relevant videos of drumming.

We provide examples where the video branch has been adapted in Fig. 6. In most cases, the retrieved titles are broadly related to the video thumbnails. However, when provided with the comments, the retrieved titles become more specific to the videos. For example, in the example from the second row of a screenshot from Mario Kart, the retrieved titles are generally about games e.g. The Castle or Sun Haven, whereas when adapting the video with the comments, the model retrieves titles specifically about Mario Kart. Similarly, in the example from the last row, the model seems to get confused about the content of the video when deprived of the comments, which provide the necessary context about feeding a fish.

Finally, in Fig. 7, we show the saliency of comments with regards to a given video and title. For this, we use the approach of masking out each comment in turn, allowing us to visualise the effect of each individual comment on the network output. We compare the output descriptor when including all comments to the descriptors with a comment masked, using the inner product as a score of similarity, and present the comments sorted from lowest to high, the expectation being that an uninformative comment will not cause a large shift in the descriptor (so will still have high similarity when excluded) whereas a salient comment will cause a larger shift (and so a lower similarity when excluded). We show results for adapting both the text branch (left) and visual branch (right), and observe that, as expected, uninformative comments such as "That was great!" and "Possibly?!?!?!?! Lol" cause little change to the descriptor, whereas comments related to objects in the video cause a larger shift. This demonstrates that the method is able to pick out and filter the relevant information.

## A.2 TRAINING DETAILS

The majority of experiments were conducted on a rented 4xA100-40GB GPU server costing approximately 170USD per day, over the course of three months. Image models (using batch size 128) and video models without comments (using batch size 50) could train on a single 40GB GPU. For TimeSformer models the visual branch was processed on a separate GPU (when training with CAM and batch size 50) or pair of GPUs (for finetuning on video benchmarks with batch size 128). Pretraining the adapter on images takes approximately one hour per epoch. Training the full video model with CAM takes approximately 6 hours per epoch. For the video experiments, we first train the CAM for 5 epochs with the backbone frozen, and then train the rest of the network for one epoch, with the backbone modified to have temporal attention as described in Section 3.3. We use the CAM with 5 comments, and adapt the visual branch of the model.

Table 9: **MSVD** Chen & Dolan (2011) Video-to-Text retrieval.

| Method | R@1 | R@5 | R@10 | MedR |
|---|---|---|---|---|
| SSB (Patrick et al., 2020b) | 21.4 | 46.2 | 57.7 | 6.0 |
| CLIP (Portillo-Quintero et al., 2021) | 59.9 | 85.2 | 90.7 | 1 |
| Ours (no comments) | 62.5 | 87.0 | 92.8 | 1 |
| Ours (no TimeSformer) | 59.1 | 85.1 | 91.0 | 1 |
| Ours | 61.6 | 87.9 | 93.4 | 1 |
| Ours (with Kinetics) | 61.3 | 87.0 | 92.4 | 1 |

Table 10: **MSRVTT** (Xu et al., 2016a) Video-to-Text retrieval.

| Method | R@1 | R@5 | R@10 | MedR |
|---|---|---|---|---|
| SSB (Patrick et al., 2020b) | 12.7 | 27.5 | 36.2 | 24.0 |
| CLIP (Portillo-Quintero et al., 2021) | 27.2 | 51.7 | 62.6 | 5 |
| Ours (no comments) | 33.0 | 57.9 | 66.9 | 4 |
| Ours (no TimeSformer) | 31.8 | 56.9 | 66.7 | 4 |
| Ours | 33.2 | 58.5 | 68.1 | 3 |
| Ours (with Kinetics) | 33.0 | 58.3 | 68.2 | 4 |

We use both photometric and temporal data augmentation. For photometric augmentation we employ random crops $(0.5 - 1.0$ scale), random horizontal flipping, and colour jitter (brightness, contrast, saturation, hue). For temporal augmentation, we first temporally subsample the input frames (which are often 30fps) according to a random stride selected uniformly from $(4, 8, 16, 32)$ and then choose

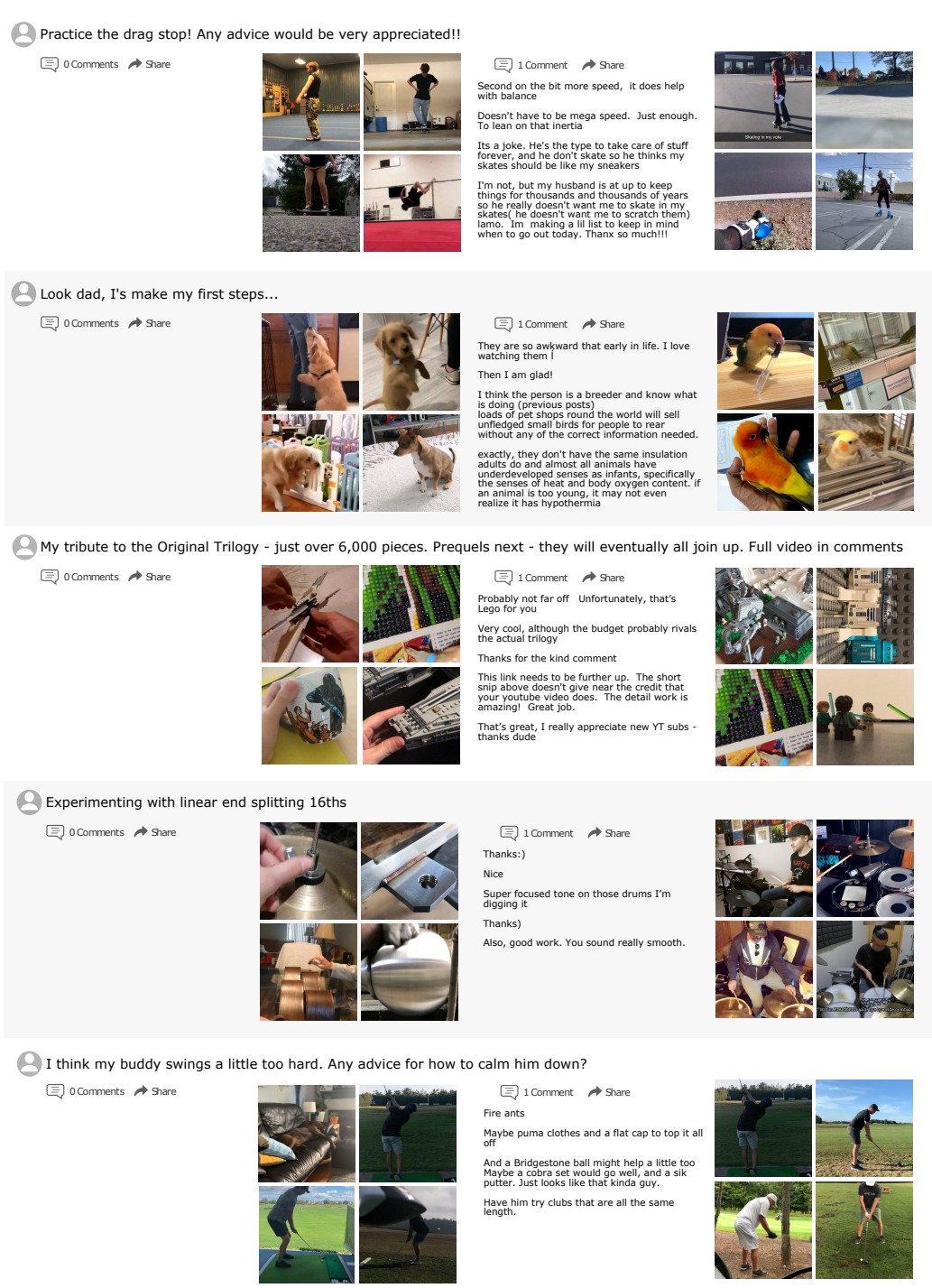

Figure 5: Examples of retrieved video thumbnails when adapting the text branch.

a random 8-frame segment uniformly. We normalise inputs using the same preprocessing as Clip (ImageNet mean and standard deviation, $224 \times 224$ input size).

At evaluation time we use a temporal stride of 16 and split the video into 8-frame chunks, taking the average of the descriptors of the chunks.

**Comments** | **Video thumbnail** | **Retrieved titles w/o adapting** | **Retrieved titles with adapting**

**Comments**

Harder the struggle, longer the snuggle.

#YOU CANT ESCAPE THE CUDDLES

"Video, video, always with the video! ...but if you must, behind the ears mom, let's go."

He seems very mellow! And he is very handsome.

I'll check your profile out

**Retrieved titles w/o adapting**

today my cat woke up more loving than usual (sorry if I wrote something wrong)

Morning routine of kneading and face smushing with my best guy ❤️

I don't know why, but my cat likes my bed so much. The first thing he does when we let him in is go to my room. He just enjoys it so much. This is his moment of happines, and mine as well. He just looks so happy!

Sound on! Luna like to purr on me in the mornings

My kitty Mordecai was tucking me in when I didn't feel well ❤️

**Retrieved titles with adapting**

Sound on! Luna like to purr on me in the mornings

Woke up with her on me

My kitty Mordecai was tucking me in when I didn't feel well ❤️

It's been a crappy week, but his cuddles definitely help sooth my soul.

I think she missed me when I was traveling. (Sound on!)

**Comments**

Thank you!

Looks really good anyway. Great work.

Just add another Vader cutaway, could be quite humorous

I'll have another look at it and see if I can make it look good

With: James Earl Jones and David Prowse As Death Vader

**Retrieved titles w/o adapting**

Why did I make this fan edit? Because of Obi-Wan.

Made another rotoscope test [oc]

My tribute to the Original Trilogy - just over 6,000 pieces. Prequels next - they will eventually all join up. Full video in comments 👍

I wanted to visualise why TROS' ending disappointed me so much, so I edited ROTJ to have an ending that's more similar to certain aspects of TROS'. Feel free to discuss.

My first attempt at video editing lol. I call this one The High Ground. Gonna want the audio on this one

**Retrieved titles with adapting**

"The Chosen One" Star Wars 3D Lenticular Fan Artwork ... created by interlacing strips in alternating orders so light reflecting at different angles shows multiple images ... hand drawn & self-produced!

Star Wars/Interstellar - My most ambitious and time-consuming project yet. Enjoy!

I wanted to visualise why TROS' ending disappointed me so much, so I edited ROTJ to have an ending that's more similar to certain aspects of TROS'. Feel free to discuss.

My tribute to the Original Trilogy - just over 6,000 pieces. Prequels next - they will eventually all join up. Full video in comments 👍

[OC] Qui-Gon & Obi Wan VS Darth Maul Except Darth Maul Fights With a 20 BLADED LIGHTSABER.

**Comments**

Rip to the guy who got blue shelled last second.

Lol thanks. Everything that could've gone right with the end of this race did. The exact opposite of some races where you go from 1st to last within seconds (here's looking at you, Moo Moo Meadows). Those green shell hits were clean

Those green shell hits were clean

Lol yeah, good point. Mk is definitely Chaos.

It is, although it's much easier in 200

**Retrieved titles w/o adapting**

Full tour of my fully decorated medieval island kingdom, 6 months of work

How it started/How it's going - Crown Trick. This is one year of progress between the playable demo and the final version of the game. Still can't believe that we are releasing the game tomorrow on Steam and Nintendo Switch

If any wonder what if when Marvelous Merchant switch places + New Location

Does the ghost in The Castle come with the DLCs or...?

One of my personal favorite parts of our game Sun Haven-- The Wishing Well

**Retrieved titles with adapting**

[MK8DX] Sometimes the Blue Shell can really help out.

[MK8DX] Is my copy of Mario Kart defective, or what?

I suck at Mario kart double dash but got 2 specials in a row (I'm best at ds)

[MKDS] Blue Shells just don't care about walls

[MK8DX] That was the funniest Mario Kart moment I've ever had (look at the Koopa)

**Comments**

Appreciate it man 😊 the vibes roll on

Sounds fire bro, keep going! Hozier rocks 🔥

exactly! Thanks

sounds great! got the birds chirping in the background just like in the recording down pretty well too

Sweet man thanks. Keep at it

**Retrieved titles w/o adapting**

It's still Friday in Hawaii! Been 2 months on my first guitar (Guild Om-150CE)... learning from Youtube

Sat to work on an old project. - it ended up as something completely different. Fingerpicking + open chords + listening to the Beatles last week. What's your thoughts?

This is my first time posting what I play on guitar, so I was pretty nervous, anyways this peace is called "Etiuda quasi tarantela".

My first post on Reddit of my singing. My take on Santeria by Sublime

Stumbled across leaves on a vine while modelling on my guitar. Probably wrong and in a different key but thought you guys might enjoy it

**Retrieved titles with adapting**

Yo yo yo this is a song I'm working on called "Lovers and Friends" - do y'all dig it?

Sat to work on an old project. - it ended up as something completely different. Fingerpicking + open chords + listening to the Beatles last week. What's your thoughts?

Sadness and Sorrow on mandolin (voice memo) - my first time letting anyone besides close friends and my teacher hear me play

My first post on Reddit of my singing. My take on Santeria by Sublime

My first post here! The song is called Cigarettes and Rain. I'm pretty new to songwriting so feedback is welcome!

**Comments**

Aww ❤️

He said thank you ❤️

Too cute

He did actually catch a couple of the pellets 😄

He wanted to catch the food :)

**Retrieved titles w/o adapting**

A reversed GIF of a post I found in r/soapmaking, link in comments.

The vibrating effect of the ripples on this pond, looked exactly like this in person. What do y'all think? Maybe not satisfying as f***, but satisfying at least?

Filmed the little piece of brutalism I printed

Slow-mo ripples in our patio mini pool

Does anyone know why my hdr is showing through my movie clip? Also when I don't use a hdri the "scene setup" option doesn't work well, the light doesn't work at all! HELP!

**Retrieved titles with adapting**

Mongo is well fed, but still wants cichlid pellets. Pterygoplichthys scrophus - rhino pleco

I dont know whats wrong with Leo today...He's acting weird and not as usual...Usually when he sees me he came and begs for food..But today i knock the glass many times to let him know im back..but he just sat at the gravel and doesn't respond to my knocks.(More in comments)

We caught Rocky eating a minnow! We were moving to an apartment so he's was in a 10 gallon but don't worry we have him in a 40 gallon at the moment 👍

Making him do tricks for his meal, might as well be a clownfish!

Caught my Jungle Vali pearling today.

Figure 6: Examples of retrieved titles when adapting the visual branch.

We randomly mask out comments with probability 0.5. We randomly skip adding the residual from the adapter with probability 0.5, which ensures that unadapted descriptors are also used in the loss and so the backbone network can still be used without the adapter.

**Title: She does this every time we feed her.**

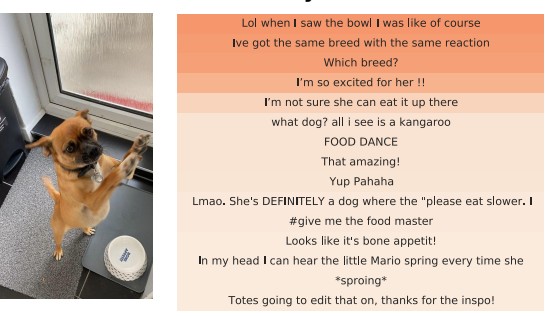
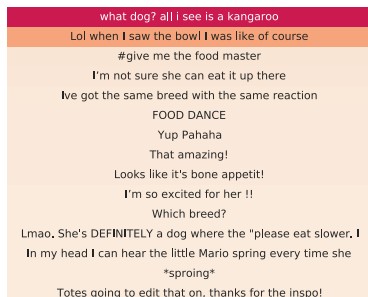

**Title: Here's Giorno's theme from JJBA.**

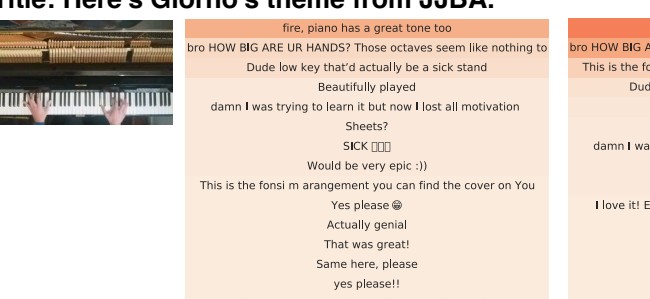

**Title: Car battery problem? No Problem!**

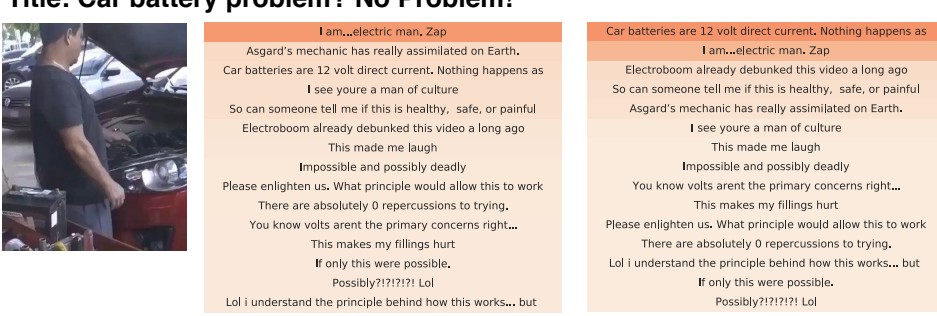

Figure 7: **Visualising comment saliency.** We show the title and thumbnail for three videos, and show the ranked saliency of comments when adapting using the Text branch (left) and Image branch (right). Comments mentioning topics relevant to the title or image are ranked highly, while irrelevant comments are lower.

All retrieval experiments are GPU accelerated using the FAISS[1] library.

## A.3    VIDEO-TO-TEXT RETRIEVAL

We report our zero-shot Video-to-Text Retrieval evaluation results on MSVD and MSR-VTT in Table 9 and  Table 10.

## A.4    KINETICS COMMENTS

In this section we will describe the details for the additional comments we retrieve for the Kinectics-700 dataset.  In Table 11 and Fig. 8 we show the distribution of the number of comments in the dataset. We collect a maximum of 10 comments and exclude videos without comments, which leaves

---

[1]https://github.com/facebookresearch/faiss

Table 11: Comments per video statistics for the KineticsComments dataset.

| #comments | 1 | 2 | 3 | 4 | 5 | 6 | 7 | 8 | 9 | 10 |
|---|---|---|---|---|---|---|---|---|---|---|
| #videos | 50322 | 21847 | 11946 | 7960 | 5596 | 4311 | 3220 | 2671 | 2245 | 1852 |

us with 111 920 videos of the originally 650 000 video clips. The majority of videos has one or two comments available. Nonetheless, our experiments show (Table 1) that our method can make use of this information to learn better representations.

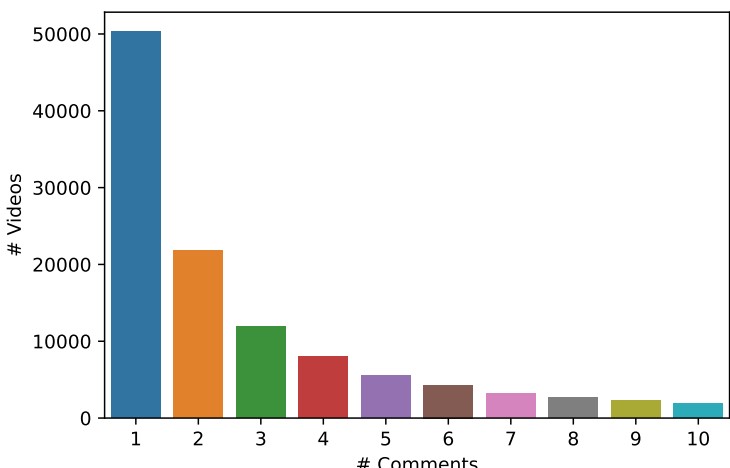

Figure 8: We show a histogram of comment statistics on KineticsComments.

### A.5 ADDITIONAL FAILURE CASES

In Fig. 9 we show additional failure cases. We find that vary vague comments "Why" or generic expressions "Ain't his fault" can distract the model from the title. In the last example, the model does not capture the concept of a sad dog due to the mention of "happy" in the comments.

### A.6 DATASET DETAILS

We report additional details on the benchmark datasets used in the paper.

**ActivityNet** ActivityNet Fabian Caba Heilbron & Niebles (2015) is a video dataset with 200 classes, 100 videos per class and 1.5 activity instances per video. The dataset contains 648h of video. There are 10,024 training videos (15,410 instances) and 4,926 validation videos (7,654 instances). The dataset itself has no license and is provided as an annotated list of youtube links. The copyright of the individual videos remains with their creators.

**MSR-VTT** MSR-VTT Xu et al. (2016b) is a video dataset with 10 categories, and 10,00 clips from 7180 videos. For training we use the 6.5k video training set with 20 captions each. It is annotated with 200,000 sentences of video descriptions created by crowd workers. The dataset itself has no license and the copyright of the individual videos remains with their creators.

**MSVD** MSVD Chen & Dolan (2011) consists of 1200/20/670 videos for training/validation/testing. The dataset itself has no license and the copyright of the individual videos remains with their creators.

**LSMDC** Large Scale Movie Description Challenge (LSMDC) Rohrbach et al. (2017) consists of 118K video clips from 202 movies. We evaluate on a test set of 1,000 videos from movies disjoint

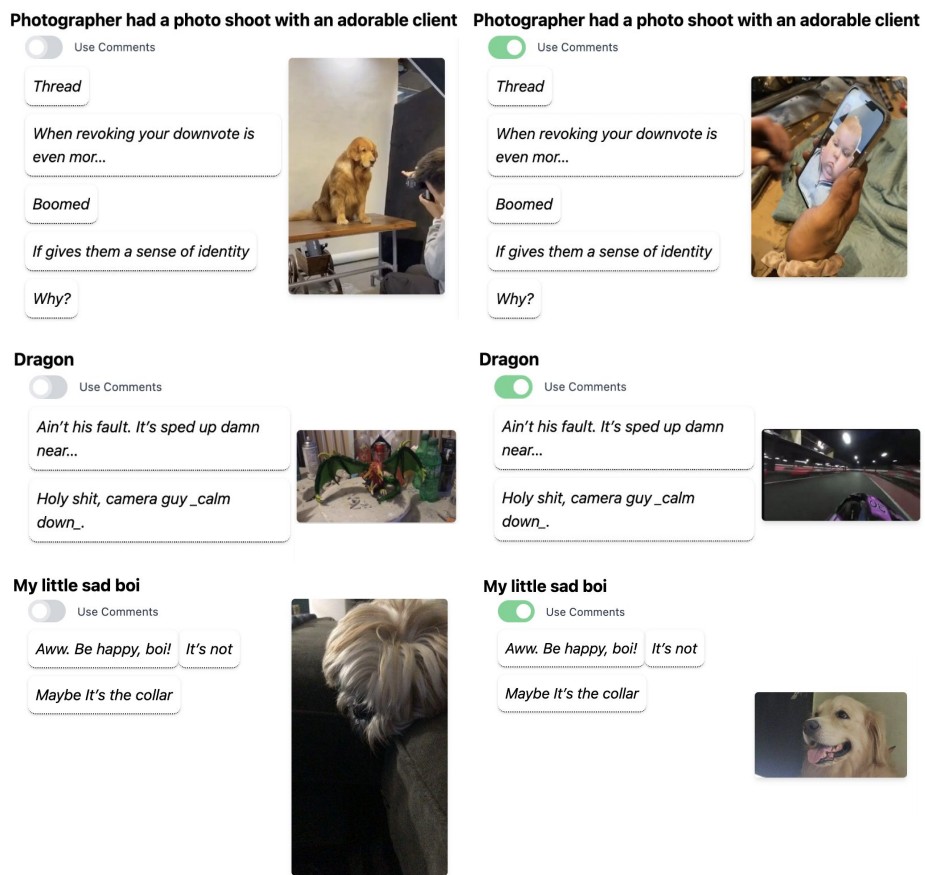

Figure 9: Examples of failure cases where using comments confounds the model and leads to a more mismatched retrieved thumbnail.

from the train and val sets, as done in (Bain et al., 2021). The dataset itself has no license and the copyright of the individual videos remains with their creators and the dataset is solely to be used for research purposes.

## A.7 DATASET CURATION

Table 12: **Prevalence of toxic text in the dataset.** We report the proportion of posts, titles, and comments that are flagged as having potentially offensive content by the open-source library Detoxify. We use a threshold of 0.9.

| Detoxify label | % titles | % comments |
|---|---|---|
| toxicity | 2.32 | 5.62 |
| severe toxicity | 0.00 | 0.00 |
| obscenity | 1.23 | 3.73 |
| identity attack | 0.00 | 0.00 |
| insult | 0.82 | 1.95 |
| threat | 0.05 | 0.07 |
| sexually explicit | 0.09 | 0.22 |

Table 13: **Bootstrapping confidence interval estimates**. Showing mean and 95% confidence intervals obtained by bootstrapping with 80% of the test sets. Recall in units of percentage points. Using models trained on RedditVC and finetuned on the respective benchmark training sets.

| Method | R@1 | R@5 | R@10 | MedR |
|---|---|---|---|---|
| MSR-VTT | 42.6±1.5 | 70.0±0.9 | 81.6±0.7 | 2.0±0.0 |
| MSVD | 48.5±1.1 | 79.2±0.7 | 87.5±0.6 | 2.0±0.0 |

Table 14: Mean recall performance and 95th percentile confidence intervals across five experiments trained with different random seeds for the best two models from Table 1 in the main paper.

| Finetune Backbone | Train. CAM | Eval. CAM | Text → Video | | Video → Text | |
|---|---|---|---|---|---|---|
| | | | R@1 | R@10 | R@1 | R@10 |
| ✗ | $g_v$ | $g_v$ | 2.65±0.09 | 12.03±0.14 | 2.79±0.16 | 12.33±0.10 |
| ✓ | $g_v$ | $g_v$ | 2.95±0.10 | 13.08±0.12 | 2.91±0.12 | 12.94±0.13 |

Table 15: We experiment with an additional modality: Audio. Our model can learn from audio as well as comments and it is even possible to combine both modalities in the same adapter module.

| Aux. modality | TVR R@1 | TVR R@10 | VTR R@1 | VTR R@10 |
|---|---|---|---|---|
| - | 2.25 | 9.85 | 2.22 | 9.28 |
| audio | 2.08 | 10.02 | 1.97 | 9.60 |
| comments | 2.24 | 10.71 | 2.01 | 10.24 |
| comments + audio | 2.62 | 12.75 | 2.30 | 12.23 |

We use the GPU implementation of the FAISS similarity search toolkit (Johnson et al., 2017) to efficiently deduplicate the dataset by indexing the video thumbnail embeddings obtained from a ResNet18. These indices are then used to discard video entries with a high similarity to other posts.

### A.8 CONFIDENCE INTERVALS

We report the mean retrieval results in Table 14 and the confidence intervals of the best two image models from the ablation study (Table 1, main paper), namely the models adapting the visual branch, with and without finetuning the backbone. These are computed across five independent training runs with different random seeds. Since training the video model takes a considerable amount of time, we estimate the variance of our method through bootstrapping the validation dataset. We generate 5 validation datasets with 80% of the full size by sampling with replacement and use the quantitative results to estimate the standard deviation, reported in Table 13.

## B ADDITIONAL EXPERIMENTS AND RESULTS

During the discussion period we have performed additional experiments based on the reviewers' suggestions.

**Additional Modality: Audio** As our context adapter module is general in terms of modality, we have performed an additional experiment by using the audio signal that comes with the video. In this experiment (see Table 15) we use the audio encoder of GDT of Patrick et al. (2021) to encode five two second audio clips and add an MLP to map the audio embedding to the CLIP feature space.

We find that using comments alone is a stronger learning signal than audio alone. Exploiting both, comments and audio, improves the performance over using each auxiliary modality alone. Using audio alone only slightly improves R10 numbers upon not using any auxiliary modality. We hypothesise this is because the embedding learned by GDT's audio encoder network is considerably different to the embedding learned by our pre-trained CLIP model. This insight is supported by the fact that fine-tuning the CLIP network together with the audio MLP is essential to learn anything in this setting.

**Ablation Image Model: Frame Time** For our image model, that operates on single frames only, we ablate the choice of frame within the video in Table 16.

We find that the choice of frame does indeed matter, with the middle frame in a video performing better than the first frame. This result is relatively intuitive since the actual content of a video usually happens in the middle of the video which is similar to the photographer bias (pictures tend to display an object in the middle of the frame).

Table 16: For the experiments using images instead of videos, we ablate the position of the frame in the video. Extracting the frame from the middle of the video is better, likely due to the fact that the important action in a video usually occurs in the middle or end of the clip.

| Model | Frame | TVR R@1 | TVR R@10 | VTR R@1 | VTR R@10 |
|---|---|---|---|---|---|
| adapting text | first | 1.36 | 7.20 | 1.45 | 7.67 |
| adapting text | middle | 1.50 | 7.46 | 1.50 | 7.97 |

Table 17: We test the robustness of the model to distractor comments (randomly sampled from the remaining dataset). We find that our context adapter module is able to deal with with bad comments better than the averaging baseline without having been trained with distractor comments.

| Model | #Distr. | TVR R@1 | TVR R@10 | VTR R@1 | VTR R@10 |
|---|---|---|---|---|---|
| averaging | 1 | 2.18 (-12.52%) | 10.50 (-10.63%) | 2.38 (-9.33%) | 11.12 (-7.85%) |
| averaging | 3 | 1.82 (-26.70%) | 8.83 (-24.80%) | 1.91 (-27.20%) | 9.47 (-21.47%) |
| averaging | 5 | 1.41 (-43.30%) | 7.64 (-34.96%) | 1.63 (-37.98%) | 8.01 (-33.60%) |
| averaging | 7 | 1.41 (-43.18%) | 6.95 (-40.83%) | 1.32 (-49.91%) | 6.66 (-44.83%) |
| adapt. title | 1 | 1.94 (-5.66%) | 9.76 (-6.92%) | 1.82 (-7.48%) | 9.28 (-7.02%) |
| adapt. title | 3 | 1.67 (-18.83%) | 8.81 (-15.96%) | 1.61 (-18.05%) | 8.42 (-15.59%) |
| adapt. title | 5 | 1.59 (-22.81%) | 8.13 (-22.46%) | 1.49 (-24.30%) | 7.81 (-21.77%) |
| adapt. title | 7 | 1.50 (-27.15%) | 7.66 (-26.95%) | 1.40 (-28.85%) | 7.19 (-27.94%) |
| adapt. img | 1 | 2.63 (-8.24%) | 12.19 (-6.71%) | 2.53 (-13.78%) | 11.87 (-9.15%) |
| adapt. img | 3 | 2.25 (-21.25%) | 10.62 (-18.71%) | 2.25 (-23.34%) | 10.62 (-18.72%) |
| adapt. img | 5 | 2.02 (-29.34%) | 9.78 (-25.13%) | 2.00 (-31.81%) | 9.53 (-27.05%) |
| adapt. img | 7 | 1.82 (-36.59%) | 8.98 (-31.29%) | 1.81 (-38.37%) | 8.83 (-32.38%) |

**Distractor Comments** In order to test the ability of our adapter to discard irrelevant information, we evaluated how our models perform when we progressively add distractor comments from other videos compared to the averaging baseline from Table 2. We find that, while both methods get gradually worse with more distractor comments, our method's retrieval performance decreases at a slower rate, as it can be seen when looking at the percentage decrease from the original score, showing that it is better equipped to discard irrelevant information. It is important to note that the model was not trained with explicit distractors and has learned this capability through the builtin attention mechanism in the context adapter model.

**Additional Dataset** To test the performance of our model on another publicly available dataset, we perform experiments using the LiveBot Dataset (Ma et al., 2019). As our pre-trained backbone is trained only in English, but the dataset is collected from a Chinese website, we translate all title and comments to English before we run our method (see Table 18).

Consistent with the results in the main paper, the retrieval performance of CLIP is surpassed by our model trained with comments. Using comments during retrieval further improves the results, confirming that comments are a useful modality and that our approach can successfully leverage this auxiliary information.

Table 18: LiveBot Dataset results. We show that even on a dataset such a LiveBt which is live comments on a video feed, our CAM can extract meaningful information from the comments.

| Model | Comments | TVR R@5 | TVR R@10 | VTR R@5 | VTR R@10 |
|---|---|---|---|---|---|
| CLIP | no | 42.0 | 50.0 | 36.0 | 44.0 |
| CLIP + CAM | no | 45.0 | 50.0 | 39.0 | 46.0 |
| CLIP + CAM | yes | 53.0 | 64.0 | 53.0 | 66.0 |

Table 19: Baseline similarity thresholding. We show that the similarity score alone cannot identify distracting comments and discarding low similarity comments decreases performance.

| Model | Sim. tresh. | TVR R@1 | TVR R@10 | VTR R@1 | VTR R@10 |
|-------|-------------|---------|----------|---------|----------|
| averaging | - | 2.33 | 11.46 | 2.59 | 11.86 |
| averaging | 0.7 | 2.11 | 10.11 | 2.35 | 10.52 |
| averaging | 0.8 | 2.16 | 9.96 | 2.20 | 9.78 |

**Filtering Comments by Similarity**  To understand the importance of *learning* to filter comments vs. using a simple thresholding technique on the similarly returned by comparing embeddings, we conduct the following experiment in Table 19. We use the similarity between a title and a comment as computed by the encoder model and remove comments that differ from the title more than a certain threshold. Since this experiments evaluates a non-learned filtering procedure we use the averaging baseline from Table 2 as reference model.

We find that removing comments generally reduces the retrieval performance as comments can indeed contain additional information. Simply thresholding the similarity metric cannot distinguish between comments that have other but useful information and comments that do not contain information related to the video. Our model learns to filter comments and can thus make use of this information.

## C  MODEL DIAGRAM

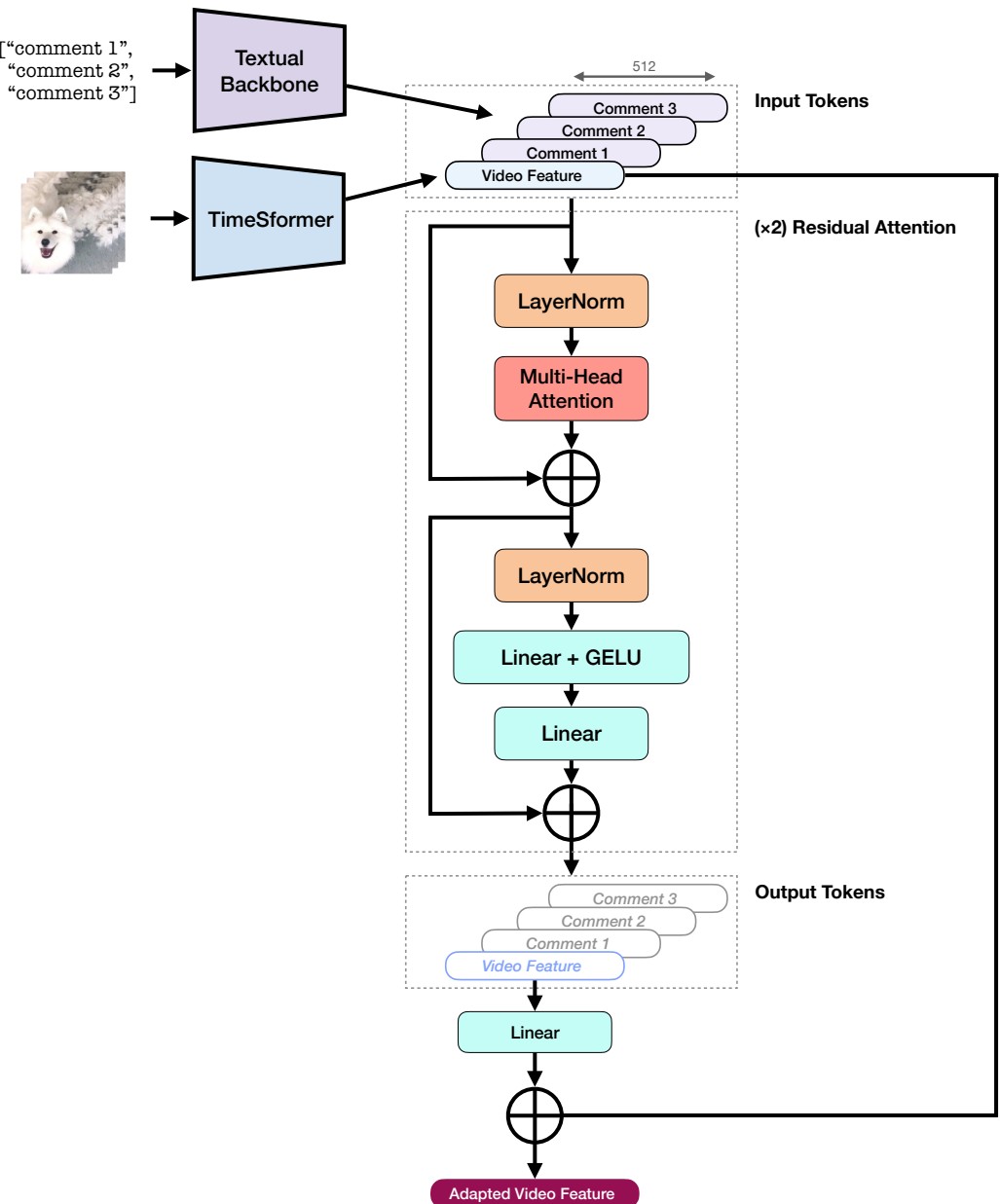

Figure 10: We show a diagram of the feature extraction and Context Adapter Module for the case of adapting the Video Feature. Multi-Head Self-Attention is performed on the input tokens (which are themselves video or textual features) as part of a transformer architecture consisting of two Residual Attention blocks. Finally the output token corresponding to the Video Feature is passed through a final linear layer and added to the original feature in a residual fashion.

