# OpenReview forum: "Learning Context-Adapted Video-Text Retrieval by Attending to User Comments"
_ICLR.cc/2022/Conference — ICLR 2022 Submitted_

### Official Review · Reviewer_SsvR · 2021-10-29

**Correctness:** 3
**Technical Novelty And Significance:** 2
**Empirical Novelty And Significance:** 3
**Recommendation:** 6
**Confidence:** 3

**Main Review:**

[Writing]
- First of all, what is "video-text retrieval"? I think this could be comprehended as a video OCR method, which apparently is not what the paper is about. According to what I have understood from the paper, I think the authors are performing "text-based video retrieval". Please reconsider the vocabulary, and also neatly define the task at the beginning of the paper.

- Table 1: What datasets do CLIP and CLIP + Our CAM use?

- References: Most references miss detailed bibliographic information. Please provide complete bibliographic information for all references.

[Evaluation]
- I am not convinced that the proposed method is superior to CLIP4Clip. Although CLIP4Clip seems to require larger-scale training, since it has already been trained and published, there should be no demerit for that. Please explain in more detail the merit of the proposed method compared to CLIP4Clip. In addition, the proposed method requires user comments, which can not always be expected in large quantity. This could actually be a demerit in some situations.

[Presentation]
- Section 4.1: Please illustrate the structure of the network.
- Fig. 3: The background color makes it difficult to see the lines, so please remove the background color, and also broaden the line widths.

**Summary Of The Paper:**

The authors propose a text-based video retrieval method based on context from weakly related user comments. Evaluation shows that considering user comments improves the retrieval performance. Compared to conventional methods, the proposed method shows similar performance with CLIP4Cilip. However, CLIP4clip needs a much larger-scale training, so the superiority of the proposed method is claimed.

**Summary Of The Review:**

The proposed method works well in case user comments are available, but even so, compared to a conventional method; CLIP4Clip, it performs similarly, so the technical merit is limited.

---

> ### Author Response · Authors · 2021-11-17
> **Response to Reviewer SsvR**
>
> We thank the reviewer for their feedback. We will clarify the raised points in detail below.
>
> ### Writing
>
> > First of all, what is "video-text retrieval"? I think this could be comprehended as a video OCR method, which apparently is not what the paper is about.
>
> “Video-Text retrieval” is a common term in multi-modal representation learning and is used in most of the cited papers. We understand why this could be a point of confusion if the reader is familiar with other literature and we will make this more clear in the revised version of the paper.
>
> > Table 1: What datasets do CLIP and CLIP + Our CAM use?
>
> Both of these are trained on our RedditVC dataset - we will make this clearer in the table.
>
> > References: Most references miss detailed bibliographic information.
>
> Thank you for spotting this. We have fixed the references and will update the paper shortly.
>
> ### Evaluation
> > I am not convinced that the proposed method is superior to CLIP4Clip. Although CLIP4Clip seems to require larger-scale training, since it has already been trained and published, there should be no demerit for that.
>
> The goal of this table is not to achieve superior performance in the no-comments context necessarily, but rather to show that we can use comments to learn good representations even when there are no comments present. CLIP4Clip does not have the ability to utilize comments at all and we show a generic method that is able to add this functionally to almost any backbone. As we show in table 1, our method can make use of comments at eval time too.
>
> We do compare against Clip4Clip and other methods in the setting of no comments to provide context and actually do find similar or better performances, despite having a much smaller pretraining dataset than Clip4Clip, which is encouraging since the strength of our model lies in exploiting comments.
>
> Our main goal was not to train a state-of-the-art video-text retrieval model, but rather to introduce our context-adapter module that is able to learn from noisy modalities such as comments (or audio as shown in our response to Reviewer 599p) and show for the first time that comments are an untapped informative modality that can help learn better, context-aware, representations.
>
> > In addition, the proposed method requires user comments, which can not always be expected in large quantity.
>
> In our evaluation we are able to show that our method can exploit the information contained in comments such that a model trained with comments performs better than a model trained without, even when there are no comments available at test time.
> Additionally, in the case without comments at test time, we are able to perform as well as existing methods (eg. CLIP4Clip and others) even though our model is operating in a different setting than it was trained in.
>
> ### Presentation
>
> > Section 4.1: Please illustrate the structure of the network.
>
> The main architecture of our context adapter model is shown in Fig. 2. The backend is CLIP extended with the TimeSformer principle [1] for which we will add a diagram to the appendix.
>
> [1] Gedas Bertasius, Heng Wang, and Lorenzo Torresani. Is space-time attention all you need for video understanding?
> arXiv preprint arXiv:2102.05095, 2021.

---

> > ### Author Response · Authors · 2021-11-23
> > **Additional Figure and Revised Paper**
> >
> > We have included a diagram of the feature extraction and Context Adapter Module for the case of adapting the Video Feature in Appendix C. Multi-Head Self-Attention is performed on the input tokens (which are themselves video or textual features) as part of a transformer architecture consisting of two Residual Attention blocks. Finally the output token corresponding to the Video Feature is passed through afinal linear layer and added to the original feature in a residual fashion.
> >
> > We have also updated the references in the revised version to a proper formatting and updated the description of Table 1.

---

> > ### Comment · Reviewer_SsvR · 2021-11-24
> > **Seems good!**
> >
> > The authors have cleared my concerns.

---

> > > ### Author Response · Authors · 2021-11-30
> > > **Great!**
> > >
> > > We are glad we have cleared your concerns! If there is nothing left, we would appreciate if you could adjust your scores after taking the rebuttal into account!

---

### Official Review · Reviewer_599p · 2021-10-31

**Correctness:** 2
**Technical Novelty And Significance:** 3
**Empirical Novelty And Significance:** 2
**Recommendation:** 5
**Confidence:** 4

**Main Review:**

1. The main idea of this paper is taking users' comments as a noisy signal to train multimodal (visual-text) video representation learning. Since the comments are not always strongly correlated to the contents that we are trying to learn, they proposed the CAM to selectively reflect the patterns discovered from the comments. Specifically, they utilize skip connections that are used in ResNet, so the model is forced to reflect patterns only if they are really necessary. This is an interesting idea and seems to work well in practice, according to the experiments.

2. This paper focuses mainly on the effect of "comments", but I think the proposed idea may be applicable more broadly. For instance, multimodal video models using audio and visual features might take advantage of similar approaches. Some videos convey more information with visual signals (e.g., the sound is edited with a background music) while some other videos convey more information with audio (e.g., a music video with the visual is the album jacket all the time). Selectively reflecting signals from each mode may be applicable to any mode, even including visual or audio, not just text (comments).

3. In this sense, this paper uses an idea which can be more generalized only within the narrow scope of "comments", and the experiments are verifying the efficacy of comments, not that of the model. For instance, the conclusion we can learn from Table 1 is that training and testing with comments help visual-text retrieval. However, this is already known. I believe the real value of this paper is coming from the model design, but that part is not well reflected in this table.

4. The authors constructed two new datasets instead of using existing publicly available ones. This significantly limits opportunities of replication by others, so I am curious if the authors are willing to publicly share the dataset. Given the scale of the two datasets, they should be valuable sources to advance this field.

5. Regarding comparison with SOTA models (in Section 4.4): The authors chose two datasets, MSVD and MSR-VTT. Although these are widely used datasets, there are many other datasets for this retrieval task. As it is known that retrieval performance highly depends on the training datasets, I'd recommend the authors to compare on other datasets like YouCook2, TVR, ActivityNet Captions, and more.

6. In Table 6, the authors are comparing the proposed models trained on CLIP + RedditVC (+K700) against baselines trained on other pre-training datasets, mostly on HowTo100M. Related to my comments #5 above, the final performance can be significantly dependent on the pre-training datasets. For instance, we observe CLIP4Clip performs comparably with the proposed model, and it looks reasonable to guess it is because all 5 rows (CLIP4Clip and 4 Ours) were pre-trained on CLIP dataset. To be comparable, pre-training datasets should be the same for all models, either on HowTo100M or CLIP, or both. It is unclear now if the improvement is coming from the novel model or from more informative pre-training datasets. (Note that the number of samples is not the only factor determining the quality of datasets. HowTo100M is relatively weaker, for example, as most clips are short and the associated text sentences are very short and less informative for many clips.) As it is hard to control all conditions across datasets, it is recommended to (pre-)train on the same datasets.

**Summary Of The Paper:**

In this paper, the authors proposed a multi-modal video representation learning and retrieval method based on it, taking advantage of visual and text embeddings, particularly users' comments on video sharing platforms. For this, the authors present a new component called Context Adapter Module (CAM), applying skip connections to selectively reflect user comments in the model. According to their experiments, the authors claim that the proposed method outperforms baseline models, by providing more contextualized representations of videos.


**Summary Of The Review:**

Overall, the paper proposes an interesting idea, which I believe to have potentially broader impact than what the current manuscript claims. I encourage the authors to conduct additional experiments to apply the idea to other modes, as well as comparing with other methods on more datasets in a fairer way.

---

> ### Author Response · Authors · 2021-11-17
> **Response to Reviewer 599P**
>
> Thank you for your thorough and constructive feedback. In the following, we address the points raised.
>
> ### Idea applicable to other modalities such as audio
>
> > 2. This paper focuses mainly on the effect of "comments", but I think the proposed idea may be applicable more broadly. For instance, multimodal video models using audio and visual features might take advantage of similar approaches.
>
> Thank you for this suggestion! We agree our context adapter should be applicable to other modalities as well and we have now conducted some preliminary experiments using audio clips, which can be seen in the table below. In this experiment we use the audio encoder of GDT [1] to encode five two second audio clips and add an MLP to map the audio embedding to the CLIP feature space.
>
> | |    Aux. modality   |   TVR R@1 |   TVR R@10 |   VTR R@1 |   VTR R@10 |
> |:---|:------------|----------:|-----------:|----------:|-----------:|
> | a) | - |      2.25 |       9.85 |      2.22 |       9.28 |
> | b) | audio  |     2.08 |      10.02 |      1.97 |       9.6   |
> | c) | comments|    2.24 |      10.71 |      2.01 |      10.24 |
> | d) | comments + audio |    2.62 |      12.75 |      2.3  |      12.23|
>
>
>
> We find that:
> - Using comments alone is a stronger learning signal than audio alone
> - Using comments and audio together improves the performance over using a single auxiliary modality
> - Using audio alone only slightly improves R10 numbers upon not using any auxiliary modality. We hypothesise this is because the embedding learned by GDT’s audio encoder network is considerably different to the embedding learned by our pre-trained CLIP model. This insight is supported by the fact that fine-tuning the CLIP network together with the audio MLP is essential to learn anything in this setting.
>
> It is important to note these are preliminary audio experiments during this discussion phase and we expect these to improve with proper hyper-parameter settings.
>
> Based on these results, we can thus show that the CAM module is indeed able to leverage information from different modalities on their own, as well as simultaneously.
>
> > 3. (...) For instance, the conclusion we can learn from Table 1 is that training and testing with comments help visual-text retrieval. However, this is already known.
>
> To our knowledge our paper is the first to show that comments can improve video-text retrieval. If we have missed a reference, we are happy to include it in the discussion and update our claims.
>
> ### Pretraining datasets
>
> > 4. The authors constructed two new datasets instead of using existing publicly available ones. This significantly limits opportunities of replication by others, so I am curious if the authors are willing to publicly share the dataset.
>
> We agree these datasets should be valuable resources and that is why we are planning to release them in a suitable form. We will soon release code for downloading comments on K400, as this dataset is already widely used. As for RedditVC, we are planning to release a subset that does not have PID to mitigate bias and privacy issues.
>
> We are currently running some experiments on the translated LiveBot dataset [2] and will report the results shortly. As our backbone network is only available in English, the translation step is required for a reasonable evaluation.
>
> > 5. (...) As it is known that retrieval performance highly depends on the training datasets, I'd recommend the authors to compare on other datasets like YouCook2, TVR, ActivityNet Captions, and more.
>
> We actually do provide more experiments where we evaluate our method on other benchmarks such as LSMDC and ActivityNet in the paper (see Tables 7-8).
>
> > 6. (...) To be comparable, pre-training datasets should be the same for all models, either on HowTo100M or CLIP, or both. It is unclear now if the improvement is coming from the novel model or from more informative pre-training datasets.
>
> We agree that evaluation in the video-text retrieval is currently less than ideal. Many models aren’t directly comparable and pretraining is not yet standardized. As many current methods are very expensive to train, we cannot level the field in this paper. However, our main goal here was not to directly compare to these methods, but to showcase how the noisy “comment-modality” is a valuable resource and can be used to learn better representations (since our models trained with and without comments are directly comparable to each other) while including the other methods to provide context.
>
> [1] Mandela Patrick, Yuki M Asano, Ruth Fong, Joao F Henriques, Geoffrey Zweig, and Andrea Vedaldi. Multimodal self-supervision from generalized data transformations. arXiv preprint arXiv:2003.04298, 2020
>
> [2] Ma, Shuming, et al. "Livebot: Generating live video comments based on visual and textual contexts." Proceedings of the AAAI Conference on Artificial Intelligence. Vol. 33. No. 01. 2019.

---

> > ### Author Response · Authors · 2021-11-23
> > **Additional Results**
> >
> > ### Livebot Dataset
> >
> > To test the performance of our model on another publicly available dataset, we perform experiments using the LiveBot Dataset as suggested by reviewer PRNC. As our pre-trained backbone is trained only in English, but the dataset is collected from a Chinese website, we translate all title and comments to English before we run our method.
> >
> > |Modality |TVR R@1|TVR R@10|VTR R@1|VTR R@10|
> > |:---|---:|---:|---:|---:|
> > |-                     |2.25     |9.85     |2.22     |9.28|
> > | audio                 |2.08     |10.02     |1.97     |9.60 |
> > | comments             |2.24     |10.71     |2.01     |10.24|
> > | comments + audio     |2.62     |12.75     |2.30   |12.23|
> >
> > Consistent with the results in the main paper, the retrieval performance of CLIP is surpassed by our model trained with comments. Using comments during retrieval further improves the results, confirming that comments are a useful modality and that our approach can successfully leverage this auxiliary information.
> >
> > We have added these new experiments and their discussions to the appendix of the paper.

---

### Official Review · Reviewer_PRNC · 2021-11-02

**Correctness:** 3
**Technical Novelty And Significance:** 3
**Empirical Novelty And Significance:** 3
**Recommendation:** 5
**Confidence:** 4

**Main Review:**

# Strengths

1. The paper is well written, and the problem is well motivated as well as clearly formulated.  The authors have made a strong case for using comments to learn better representations. They have adequately explained the challenges of using this modality, which explains why it was not used in earlier work. Although I was not aware of research in this area, I was able to follow the paper clearly.
2. As far as I can tell, the idea of using comments while ignoring irrelevant comments has not been tried before. Hence, having an attention mechanism that can actually ignore irrelevant comments can aid many machine learning tasks, including representation learning.
3. The experiments are carried out in diverse settings, which is impressive to carry out. However, I would expect this method to be tried on a public video dataset that already has comments. The dataset released by [1] can be a good fit.
4. I appreciate that the limitations of this work are properly addressed. However, as shown in Figure 4, it concerns me that most real-world videos will have “distractor” comments which will be relevant to some part of the images in the video and may result in noisy representations.

# Weaknesses

1. It is not clear if something complicated like the Context Adapter Module is required to discard v_i and t_i pairs, where v_i and t_i are not equally informative of each other. Can a simple heuristic be used instead? For instance, if the initial value of A_{ij} does not qualify a certain threshold, then they are probably NOT equally informative of each other and are worth discarding.
2. Similarly, can a heuristic be used to discount off-topic comments that have a low similarity score with the title? It would be great if the authors can compare their method with a heuristic like this.
3. In the experimental section under 4.3, the authors show that varying the number of comments leads to improvement in performance. However, to simulate a realistic scenario, a subset of these comments should be unrelated or ambiguous. As an ablation study, the ratio of these unrelated comments should also be varied. It is difficult to conclude if the attention mechanism is working in absence of these experiments.
4. The authors mention that they collected comments from the Kinetics dataset. For realistic use cases, it would be great to see experiments on a dataset which has video lengths larger than 10 seconds. Moreover, it would be important to see some statistics on the Kinetics dataset (particularly a histogram for number of comments on different videos). The authors mention that there are at least 3 comments, but more statistics would be appreciated. This is important because this is the only video dataset with comments that will be publicly released by the authors.

# Other Concerns
1. The majority of the experiments involve using the RedditVC dataset. The authors mention that they do not plan to release this dataset due to dataset bias issues. This makes reproducibility a major concern. It would be ideal if the authors can evaluate their method on a publicly available video dataset with comments [1].

2. It would be important to add a significance test for these experiments. There are no error bars reported in the experiments. Were the experiments run multiple times to estimate errors in retrieval performance (particularly in Table 1)?

3. While referring to the Appendix, it would be great if the authors can link to a particular section. For example, in Sec 4.5, Limitations, the reader would want to know where exactly they can look for additional examples, rather than simply mentioning appendix. I spent quite some time scrolling through the appendix to find more examples of limitations but could not find any.

# References
[1] Ma, Shuming, et al. "Livebot: Generating live video comments based on visual and textual contexts." Proceedings of the AAAI Conference on Artificial Intelligence. Vol. 33. No. 01. 2019.

Edit after response:
Thank you for your detailed response.
I would like to agree again that the proposed method has some merit. However, my main concern about the applicability of this module for realistic scenarios where the number of comments is very large still remains. I can see that this method will work when there are only a handful of diverse comments. It is unclear if it would work in a setting with large number of comments, where the number of distractor comments is larger than the non-distractor ones.

**Summary Of The Paper:**

This paper proposes the use of user comments in addition to videos and titles to learn better representations for retrieval. Since user comments may be loosely related to the video, they use an attention-based mechanism to ignore irrelevant user comments. Their experiments show that using comments using this mechanism leads to better contextualized representations, which lead to competitive results on standard benchmarks.

**Summary Of The Review:**

Please refer to the discussion in the Main Review.

---

> ### Author Response · Authors · 2021-11-17
> **Response to Reviewer PRNC**
>
> We thank the reviewer for their comments and suggestions! We are running the suggested experiments to provide further support of our method and will respond to the review in detail below.
>
> ### W1. Thresholding the similarity instead
>
> > It is not clear if something complicated like the Context Adapter Module is required to discard v_i and t_i pairs, where v_i and t_i are not equally informative of each other (...) For instance, if the initial value of A_{ij} does not qualify a certain threshold, then they are probably NOT equally informative of each other and are worth discarding.
>
> We understand the idea of directly thresholding the similarity to detect unhelpful samples, but we are unsure how that would fit into our framework using comments. Further clarification of the suggested baseline would be helpful. We can filter the dataset before training with this, or filter during/after training with a simple double contrastive loss. However, the final model would not be able to process more than a (video,title) pair. How would the resulting model make use of the comments?
>
> ### W2. Thresholding the similarity for comments
>
> We are running this baseline experiment and will report the results as soon as they are ready.
>
> A benefit of the adapter module is that it allows informative comments to be leveraged even when the title is irrelevant or ambiguous (eg “I found this today”), so we believe that the thresholding approach based on title/comment similarity alone would not capture this sort of scenario.
>
> ### W3. Evaluation with distractor comments
>
> > In the experimental section under 4.3, the authors show that varying the number of comments leads to improvement in performance. However, to simulate a realistic scenario, a subset of these comments should be unrelated or ambiguous. As an ablation study, the ratio of these unrelated comments should also be varied.
>
>
> Thank you for the suggestion! We posit that increasing the number of comments results in naturally increasing the number of irrelevant or ambiguous comments, as it can be seen when looking at a few examples in our supplementary material.
> We are currently running an evaluation where we add more and more distractor comments (random comments from other videos). We will measure the robustness of the retrieval wrt. to the number of distractors and will report results as soon as they are ready.
>
> ### W4. Kinetics comments dataset
>
> >  For realistic use cases, it would be great to see experiments on a dataset which has video lengths larger than 10 seconds.
>
> Our RedditVC dataset does have many videos over 10 seconds (the average length is 33s). The retrieval benchmark datasets such as MSRVTT also contain videos longer than 10 seconds.
>
> > Moreover, it would be important to see some statistics on the Kinetics dataset (particularly a histogram for number of comments on different videos).
>
> We will add all dataset details (number and length of comments per video, histogram, etc.) for KineticsComments to the paper and post a revised version soon.
>
> ### Minor concerns
>
> > The majority of the experiments involve using the RedditVC dataset. The authors mention that they do not plan to release this dataset due to dataset bias issues.
>
> We do plan to release a subset of the RedditVC dataset after removing PID and mitigating bias and privacy issues. We also plan to release code for downloading the comments on KineticsComments.
>
> #### Livebot Dataset
>
> Thank you very much for the reference! We were not previously aware of any video dataset with comments. We have now downloaded this dataset, although we have run into some issues with the data preparation (as described in [1] “lack of video mapping between the raw dataset and the processed dataset” - but we shall use the version from [1]).
>
> The other difficulty is that the comments and titles are in Chinese, while our model is based on CLIP which is pretrained on an English corpus. We have used machine translation to translate 100 test set videos and 5 comments per video, but we cannot easily translate the full dataset. The translations seem to be of mixed quality so this may affect the results, but we will endeavour to provide some meaningful retrieval evaluation results.
>
>
> #### Significance test
>
> We report confidence intervals obtained from 5 training runs with different random seeds on RedditVC in table 13.  For the full video models, which are compute intensive to train, we instead use bootstrapping with 80% of the test sets in table 12 on MSR-VTT and MSVD in section A.6.
>
> #### More precise references
>
> Thank you! We will update the paper with more precise references to the appendix. Additional examples of our method can be found in the appendix in Fig. 5-7 and in the supplementary material, however, we will make sure to highlight other potential limitations in the revised paper.
>
> [1] Wu, H., Jones, G. J., & Pitie, F. (2020). Response to LiveBot: Generating Live Video Comments Based on Visual and Textual Contexts. arXiv

---

> > ### Author Response · Authors · 2021-11-23
> > **Additional Results**
> >
> > Please find below our results for the additional experiments.
> >
> > ### Thresholding the similarity for comments
> >
> > After manually choosing suitable thresholds for the similarity, we tried discarding comments that were dissimilar to the title when finetuning the averaging baseline model from Table 2, row f. We found that filtering comments by a similarity threshold negatively impacts the performance, which confirms our hypothesis that comments can add new information that would not be found in the title.
> >
> > |Model | sim. thresh.|TVR R@1|TVR R@10|VTR R@1|VTR R@10|
> > |:---|:---|---:|---:|---:|---:|
> > |averaging | - |     2.33 |     11.46 |     2.59 |     11.86 |
> > |averaging | 0.7 |     2.11 |     10.11 |     2.35 |     10.52 |
> > |averaging | 0.8 |     2.16 |      9.96 |     2.20 |      9.78 |
> >
> > ### Evaluation with distractor comments
> >
> > For this experiment, we added progressively more irrelevant comments from other videos and tested both our method and the averaging baseline.
> >
> > We find that, while both methods get gradually worse with more distractor comments, our method’s retrieval performance decreases at a slower rate, as it can be seen when looking at the percentage decrease from the original score, showing that it is better equipped to discard irrelevant information. It is important to note that the model was not trained with explicit distractors and has learned this capability through the builtin attention mechanism in the context adapter model.
> >
> > |Model | num. distractors|TVR R@1|TVR R@10|VTR R@1|VTR R@10|
> > |:---|:---|---:|---:|---:|---:|
> > |averaging | 1 |   2.18 (-12.52\%) |  10.50 (-10.63\%) |   2.38 (-9.33\%) |   11.12 (-7.85\%)|
> > |averaging | 3 |    1.82 (-26.70\%) |   8.83 (-24.80\%) |  1.91 (-27.20\%) |   9.47 (-21.47\%)|
> > |averaging | 5 |    1.41 (-43.30\%) |   7.64 (-34.96\%) |  1.63 (-37.98\%) |   8.01 (-33.60\%) |
> > |averaging | 7 |    1.41 (-43.18\%) |   6.95 (-40.83\%) |  1.32 (-49.91\%) |   6.66 (-44.83\%)|
> > |adapt. title | 1 |    1.94 (-5.66\%) |    9.76 (-6.92\%) |   1.82 (-7.48\%) |    9.28 (-7.02\%) |
> > |adapt. title | 3 |   1.67 (-18.83\%) |   8.81 (-15.96\%) |  1.61 (-18.05\%) |   8.42 (-15.59\%) |
> > |adapt. title | 5 |   1.59 (-22.81\%) |   8.13 (-22.46\%) |  1.49 (-24.30\%) |   7.81 (-21.77\%) |
> > |adapt. title | 7 |   1.50 (-27.15\%) |   7.66 (-26.95\%) |  1.40 (-28.85\%) |   7.19 (-27.94\%)  |
> > |adapt. img | 1 | 2.63 (-8.24\%) |   12.19 (-6.71\%) |  2.53 (-13.78\%) |   11.87 (-9.15\%) |
> > |adapt. img | 3 | 2.25 (-21.25\%) |  10.62 (-18.71\%) |  2.25 (-23.34\%) |  10.62 (-18.72\%) |
> > |adapt. img | 5 |   2.02 (-29.34\%) |   9.78 (-25.13\%) |  2.00 (-31.81\%) |   9.53 (-27.05\%) |
> > |adapt. img | 7 |   1.82 (-36.59\%) |   8.98 (-31.29\%) |  1.81 (-38.37\%) |   8.83 (-32.38\%) |
> >
> >
> > ### Livebot Dataset
> >
> > To test the performance of our model on another publicly available dataset, we perform experiments using the LiveBot Dataset as suggested. As our pre-trained backbone is trained only in English, but the dataset is collected from a Chinese website, we translate all title and comments to English before we run our method.
> >
> > |Modality |TVR R@1|TVR R@10|VTR R@1|VTR R@10|
> > |:---|---:|---:|---:|---:|
> > |-                     |2.25     |9.85     |2.22     |9.28|
> > | audio                 |2.08     |10.02     |1.97     |9.60 |
> > | comments             |2.24     |10.71     |2.01     |10.24|
> > | comments + audio     |2.62     |12.75     |2.30   |12.23|
> >
> > Consistent with the results in the main paper, the retrieval performance of CLIP is surpassed by our model trained with comments. Using comments during retrieval further improves the results, confirming that comments are a useful modality and that our approach can successfully leverage this auxiliary information.
> >
> > We have added these new experiments and their discussions to the appendix of the paper. Additionally, in Appendix A.4 we have added an overview of KineticsComments including the dataset statistics and more qualitative failure cases in Appendix A.5 (we have also updated the Limitations section to refer to the precise location in the appendix).

---

> ### Author Response · Authors · 2021-11-29
> **Response to updated review**
>
> > Edit after response: Thank you for your detailed response. I would like to agree again that the proposed method has some merit. However, my main concern about the applicability of this module for realistic scenarios where the number of comments is very large still remains. I can see that this method will work when there are only a handful of diverse comments. It is unclear if it would work in a setting with large number of comments, where the number of distractor comments is larger than the non-distractor ones.
>
> Thank you for your update!
>
> Firstly, we would like to emphasise that our scenario is indeed realistic, as we download real online data without filtering (except for harmful content) and can show that our method learns better representations from comments than without. We show this on 3 datasets with comments (RedditVC, KineticsComments and LiveBot) and 4 retrieval benchmarks (MSR-VTT, ActivityNet, LSMDC, MSVD).
>
> Secondly, all of the 3 datasets with comments have videos that have a much larger number of comments than the fixed numbers we have used in our experiments. However, as our results show, it is enough to only use a subset of these to see significant gains, with diminishing returns when using larger numbers (as Figure 3 shows). In order to still take advantage of the videos with larger number of comments, we randomly sample the comments during training, hence, the model gets to see more than the set number at train time. During evaluation, we select the first n comments, where the order corresponds to the order they appear online, which is often in order of relevance or popularity.
>
> Although distractor comments are indeed a limitation of our method, we have shown that our adapter is better able to cope with them than the other baselines. As the ratio of relevant to irrelevant comments should be similar at larger numbers as well as smaller, we have little reason to believe our method would not work. Nevertheless, as this is the first method that shows that comments can be valuable in video-text retrieval, we agree this is worth investigating in future work.
>
> In the replies we also confirm that the adapter works with audio clips instead of comments. All evaluations are performed on real, large-scale datasets.
> We hope this addresses your concern!

---

### Official Review · Reviewer_2J7x · 2021-11-04

**Correctness:** 2
**Technical Novelty And Significance:** 2
**Empirical Novelty And Significance:** 2
**Recommendation:** 6
**Confidence:** 3

**Main Review:**

Strengths of the paper:

S1. The idea is intuitive.  Indeed a lot of videos now have user comments.  This is especially true for popular videos.  Having said that, popular videos probably are already getting a lot of signals from the popularity alone (likes, votes, etc.).  I'd be curious how the method could help retrieve less popular but relevant videos, where there may be some but not a lot of user comments in the first place (as it's a less popular video).

S2. The method itself is straightforward.

S3. Experiments are conducted on real datasets.  There is also ablation type of experiments looking into various components of the model. However, it is not immediately clear if there's a baseline that just throws all the comments as a single piece of text (just like a title would be), without attention to filter them out.  If so, that would be a more comparable baseline as it has the same information, but with a simpler processing.

Weaknesses of the paper:

W1. The novelty is rather incremental.  It's an extension of CLIP that already has a notion of video and text (e.g., title). The extension is in terms of user comments, which is another form of the text.  Though seemingly useful, the step to include comments as another form of text is not that surprising technically.

W2. Though the paper emphasises heavily on the video element, page 6 states that experiments are conducted on a single still frame.  There needs to be more discussion on the generalisability of the result.  I wonder if the choice of frame may affect the overall results.

W3.  The treatment of user comments as another piece of text is rather simplistic.  There could be richer structures to exploit, for instance the response/reply or discussion thread nature of some of these comments, which would be more interesting given the nature of this data.


**Summary Of The Paper:**

The paper looks into video-text retrieval problem.  The claim is that existing works mostly rely on titles and captions.  The paper argues for using user comments.  The challenge is that not all user comments are meaningful or relevant.  Therefore, the paper looks into attention mechanism to filter out the irrelevant content. The main contribution is a context adapter module based on transformer that allows relating visual input and textual input. The base architecture seems to have been derived from CLIP.

**Summary Of The Review:**

The paper proposes an intuitive idea of leveraging on user comments to improve video-text retrieval, but the technical novelty is rather incremental.

---

> ### Author Response · Authors · 2021-11-17
> **Response to Reviewer 2J7x**
>
> We thank the reviewer for their comments, suggestions and positive evaluation of the paper. Please find detailed responses below.
>
> ### Combining comments into one text
>
> > S3. However, it is not immediately clear if there's a baseline that just throws all the comments as a single piece of text (just like a title would be), without attention to filter them out. If so, that would be a more comparable baseline as it has the same information, but with a simpler processing.
>
> We agree that combining all comments into one input text seems like a natural baseline, however, this would be limited by the maximum sequence length of the pretrained text encoder model, which in CLIP's case is 77. Additionally, we do show baselines with simpler preprocessing such as averaging and random swap in Table 2. We did try concatenating the features themselves but the averaging baseline proved to be better.
>
>
> ### W1. Novelty
>
> > The novelty is rather incremental. It's an extension of CLIP that already has a notion of video and text (e.g., title). The extension is in terms of user comments, which is another form of the text. Though seemingly useful, the step to include comments as another form of text is not that surprising technically.
>
> Although our method uses CLIP as a backbone since it is a natural fit for this task, it is not exclusive to CLIP and should be easily applicable to other models as well.
>
> We agree that leveraging comments is an intuitive idea, however, training with comments in practice is not straightforward since they’re often only weakly correlated with the video and can even hurt performance if used naively. Hence, the technical novelty is in introducing a method that is able to make use of the valuable information present in comments, while discarding the irrelevant ones.
>
> To summarise, our main technical contribution is two-fold: 1) introducing our proposed context-adapter module that is able to learn from noisy modalities, and 2) demonstrating for the first time that user comments can help video-text retrieval.
>
>
> ### W2. Experiments only on images
>
> > Though the paper emphasises heavily on the video element, page 6 states that experiments are conducted on a single still frame.
>
> We do use video. The statement on page 6 refers only to the ablation experiments in Tables 2, 3 and 5. In Table 1 we show that using video does indeed help to learn better representations and Tables 5,6,7,8 and the supplement evaluate our video models.
> For the single image experiments, we are now running a comparison between evaluating on the first vs. the middle frame of a video and will post the results as soon as they are ready.
>
> ### W3. Exploit the richer structure of comments
>
> > The treatment of user comments as another piece of text is rather simplistic. There could be richer structures to exploit, for instance the response/reply or discussion thread (...)
>
> We fully agree that there is potentially more information in the comment modality that could be leveraged in future work, such as the hierarchical nature of comment threads.
>
> However, for our experiments, we decided to use multiple top-level comments without including the replies since this would provide the most diverse signal without overwhelming the adapted feature.
>
> Having already shown that comments can improve the learned representations and retrieval performance in this paper, we hope that the community will start to pay attention to this modality and find even better ways to make use of it.

---

> > ### Author Response · Authors · 2021-11-23
> > **Additional Results**
> >
> > Thank you for suggesting to evaluate the choice of frame for the experiments on images instead of videos.
> >
> > |Model | Frame|TVR R@1|TVR R@10|VTR R@1|VTR R@10|
> > |:---|:---|---:|---:|---:|---:|
> > |adapting text | first |     1.36 |      7.20 |     1.45 |      7.67 |
> > |adapting text | middle |     1.50 |      7.46 |     1.50 |      7.97 |
> >
> > We find that the choice of frame does indeed matter, with the middle frame in a video performing better than the first frame. This result is relatively intuitive since the actual content of a video usually happens in the middle of the video which is similar to the photographer bias (pictures tend to display an object in the middle of the frame). Using videos is still better than choosing the middle frame.
> >
> > We have added this experiment and the discussion to the appendix of the updated paper.

---

### Author Response · Authors · 2021-11-23
**Revision Summary**

Dear reviewers, thank you for your time and the constructive reviews. Based on the questions in the reviews, we have added additional discussions to our paper and conducted five new experiments to validate our replies.

We have responded to the individual comments below and have updated the paper correspondingly. The new results and analysis are found in the appendix (Appendix B and C) on pages 18-21.

The main changes are:

* An experiment with audio shows that our context adapter also works for this new modality.
* We show that our method is more robust to distractor comments than the baselines.
* We use the public LiveBot dataset to show retrieval performance on another benchmark.
* The choice of frame in the video does matter for our experiments with single images.
* We add another baseline of filtering comments based on similarity and find that it does not work well, as the similarity score cannot distinguish well between comments that have other information from comments that are irrelevant.
* We have added a model figure.
* Updated the paper to include the new results and some additional information on dataset details, more failure cases and small other changes (marked in red).

---

### Decision · Program_Chairs · 2022-01-20

**Decision:**

Reject

**Comment:**

This paper focuses on how to improve video-text retrieval via using additional user comments, and uses an attention mechanism to filter out the irrelevant comments. The main contribution is a context adapter module that allows learning from the auxiliary modality through an attention mechanism. The reviewers appreciated the overall idea's intuition and well-written paper, but they also felt that the technical novelty is incremental, and that the treatment of user comments should be more intuitive via the dialogue thread structure. There were also concerns about the applicability of the context adapter module to more realistic scenarios with much longer videos, where the number of comments is very large, and where number of distractor comments is larger than the non-distractor ones.